

# Tree-grass phenology information improves light use efficiency modelling of gross primary productivity for an Australian tropical savanna

Caitlin E Moore[1], Jason Beringer[1,2], Bradley Evans[3,4], Lindsay B Hutley[5], Nigel J Tapper[1]

[1] School of Earth, Atmosphere and Environment, Monash University, Clayton, VIC, 3800, Australia

[2] School of Earth and Environment, The University of Western Australia, Crawley, WA, 6009, Australia

[3] Department of Environmental Sciences, The University of Sydney, Eveleigh, NSW, 2015, Australia

[4] Terrestrial Ecosystem Research Network Ecosystem Modelling and Scaling Infrastructure, The University of Sydney, Eveleigh, NSW, 2015, Australia

[5] School of Environment, Research Institute for the Environment and Livelihoods, Charles Darwin University, Casuarina, NT, 0909, Australia

*Correspondence to*: Caitlin E Moore (caitlin@moorescience.com.au)

## Abstract

The coexistence of trees and grasses in savanna ecosystems results in marked phenological dynamics that vary spatially and temporally with climate. Australian savannas comprise a complex variety of life forms and phenologies, from evergreen trees to annual/perennial grasses, producing a boom-bust seasonal pattern of productivity that follows the wet-dry seasonal rainfall cycle. As the climate changes into the 21st Century, modification to rainfall and temperature regimes in savannas is highly likely. There is a need to link phenology cycles of different species with productivity to understand how the tree-grass relationship may shift in response to climate change. This study investigated the relationship between productivity and phenology for trees and grasses in an Australian tropical savanna. Productivity, estimated from overstory (tree) and understory (grass) eddy covariance flux tower estimates of gross primary productivity (GPP), was compared against two years of repeat time-lapse digital photography (phenocams). We explored the phenology-productivity relationship at the ecosystem scale using moderate resolution imaging spectroradiometer (MODIS) vegetation indices and flux tower GPP. These data were obtained from the Howard Springs OzFlux/Fluxnet site (AU-How) in northern Australia. Two greenness indices were calculated from the phenocam images; the green chromatic coordinate (GCC) and excess green index (ExG). These indices captured the temporal dynamics of the understory (grass) and overstory (trees) phenology, and were mostly well correlated with tower GPP for understory ($r^2$ = 0.65 to 0.72) and overstory ($r^2$ = 0.09 to 0.23). The MODIS enhanced vegetation index (EVI) correlated well with GPP at the ecosystem scale ($r^2$ = 0.70).





Lastly, we used GCC and EVI to parameterise a light use efficiency (LUE) model and found it to
improve the estimates of GPP for the overstory, understory and ecosystem. We conclude that
phenology is an important parameter to consider in estimating GPP from LUE models in savannas and
that phenocams can provide important insights into the phenological variability of trees and grasses.
**Key Words**
Eddy covariance, phenocam, leaf area index, photosynthetically active radiation, light use efficiency,
MODIS, OzFlux
**1 Introduction**
Savanna ecosystems are defined by the coexistence of trees and grasses, and have evolved to
dominate one fifth of the terrestrial land surface (Scholes and Archer, 1997;Grace et al., 2006). In
tropical savanna, trees utilise the $C_3$ photosynthetic pathway, whereas the grasses use the more
recently evolved $C_4$ pathway, being more efficient at taking up carbon in hot environments with
limited water and nutrient availability (Sage, 2004;Osborne and Beerling, 2006). Savannas are
typically found in wet/dry climates that over time have shaped the tree-grass structure and phenology
seen today. Fire also plays a role in shaping savanna phenology and structure, with recurrences often
every 1-5 years (Hoffmann et al., 2012;Beringer et al., 2015), fire consumes cured grass biomass in
the dry season and supresses growth of juvenile overstory species, resulting in a range of plant
phenology responses to deal with it (Bond, 2008;Murphy et al., 2010;Werner and Franklin, 2010).
Herbivory, drought and land-use change are additional disturbances that commonly occur in savannas
(Hutley and Beringer, 2011). These complex interactions are believed to be the primary reason for the
co-dominance of trees and grasses in savanna ecosystems, as well as for the phenological variability
displayed (Bond et al., 2003;Van Langevelde et al., 2003;Bond, 2008;Hanan and Lehmann,
2010;Lehmann et al., 2014).
The climate and disturbance regime in savannas plays an important role in shaping plant phenology.
$C_4$ savanna grasses typically follow a boom-bust phenological cycle, where they rapidly produce
biomass in the wet season and display an annual or perennial die-back phenology in the dry season
(Bond, 2008;Ratnam et al., 2011). $C_3$ savanna trees, in contrast, can range from having a fully
deciduous phenology to remaining evergreen throughout the dry season. In Australian savannas, the
understory is dominated by $C_4$ annual grasses with a small portion represented by juvenile overstory
species (Werner and Franklin, 2010;Werner and Prior, 2013) and perennial grasses. Evergreen
eucalypt species make up the bulk (~ 80 %) of the overstory in Australian savannas (Hutley et al.,
2011), however, semi-, brevi- and fully deciduous species are found to a lesser degree throughout
(Williams et al., 1997) and contribute to the seasonal fluctuation of canopy leaf area (O'Grady et al.,
2000;Whitley et al., 2011). Tree-grass ratios are driven by annual rainfall, and in Australia there is a



strong rainfall gradient from the coast inland (Rogers and Beringer, 2016), resulting in northern high
rainfall (mesic) regions supporting higher tree-grass ratios and drier southern (xeric) regions
supporting higher grass-tree ratios (Hutley et al., 2011;Ma et al., 2013).
The monitoring of savanna phenology can inform how savannas might respond to climate change. At
the regional scale, the timing of phenological events varies widely for savannas due to variability in
the occurrence and duration of rainfall events (Ma et al., 2013). Phenology, in turn, influences the
productivity and growth (carbon cycle) of an ecosystem, as well as its water and nutrient cycles
(Noormets, 2009;Richardson et al., 2013). The savanna region of Australia is projected to experience
warming and increased rainfall (variability and amount) under climate change (Reisinger et al., 2014),
which is likely to impact savanna phenology and its interactions with the carbon, nutrient and water
cycles (Kanniah et al., 2010;Scheiter et al., 2015). There is a need for better understanding of what
governs savanna phenology in order to predict how it may be affected by climate change (Beringer et
al., 2016a).
Due to the large extent and spatial variation of savannas, satellite remote sensing provides a useful
tool (Broich et al., 2015) for examining the interactions of savanna phenology with productivity.
Vegetation indices such as the normalised difference vegetation index (NDVI) (Tucker, 1979) and
enhanced vegetation index (EVI) (Huete et al., 2002) provide valuable measures of savanna
phenological variability from the landscape to global scale (Ma et al., 2013;Ma et al., 2014). Likewise,
the MODIS gross primary productivity (GPP) product (MOD17 A2/A3, Running and Zhao, 2015)
offers the most reliable means of estimating large scale savanna productivity (Grace et al., 2006;Ryu
et al., 2011), but has been shown to underestimate savanna GPP, particularly during the transition
between the wet and dry seasons (Kanniah et al., 2009;Whitley et al., 2011;Ma et al., 2014). Core
issues surrounding the remoteness of satellite sensors, the effects of cloud contamination on daily data
collection, the diffuse nature of light and the need to aggregate imagery spatially and temporally for
contiguous scenes, results in coarse temporal resolution (i.e. 8 or 16 day) satellite data products that
can be problematic for identifying change in seasonally cloudy tropical environments (Eberhardt et al.,
2016) where rapid (i.e. 1-2 weeks) phenological change is common (Williams et al., 1997;Moore et
al., 2016b).
A novel approach to alleviate some of the limitations of satellite remote sensing is to use *in situ*
automated time-lapse cameras (phenocams) that can collect high temporal resolution (hourly to daily)
images of vegetation within and above an ecosystem (Richardson et al., 2007;Hufkens et al.,
2012;Sonnentag et al., 2012;Moore et al., 2016b). The proximity of these cameras to ecosystem
vegetation allows them to capture important information about vegetation cover change via leaf
emergence and senescence (Richardson et al., 2007;Richardson et al., 2009a;Keenan et al., 2014) that
can be linked with measures of ecosystem GPP (Tans et al., 1990;Richardson et al., 2010;Toomey et



al., 2015). Phenocam data have also been used for parameterising light use efficiency (LUE) models
(in a similar way to MODIS GPP) that describe ecosystem GPP through the relationship of absorbed
photosynthetically active radiation (APAR) with that of plant LUE (Migliavacca et al., 2011).
In this study, we aim to contribute a detailed assessment of phenological cover change, and its
relationship with productivity, for a mesic tropical savanna in northern Australia over 2 years. Our
objectives are to (i) determine the utility of phenocams for identifying change in overstory and
understory vegetation greenness; (ii) quantify the relationship between savanna overstory and
understory phenology and productivity on seasonal and annual timescales; (iii) test if phenocam
indices can be used as a proxy for improvement of a LUE model that is widely used to estimate GPP;
and (iv) test the applicability of MODIS EVI for improving estimates of ecosystem scale GPP. To do
this we utilise one of the first phenocam datasets obtained in Australian ecosystems, along with
MODIS EVI, and couple them with previously collected ecosystem, overstory and understory eddy
covariance data (Moore et al., 2016a) to tease apart the tree and grass phenology-productivity
relationship in Australian savanna.

## 2 Methods

To address each of our objectives, we used a combination of eddy covariance and phenocam imagery
along with information about overstory leaf area index (LAI), understory biomass and the radiation
use of the overstory, understory and ecosystem over time. These data were used to tease apart the
relationship between productivity and phenology for the trees (overstory) and grass (understory) so
we could identify how they varied throughout the 2 year study period. Phenocam greenness
phenology information and MODIS EVI were also used to parameterise a LUE model that we then
used to estimate overstory, understory and ecosystem GPP.

### 2.1 Site Description

This study was conducted at the Howard Springs OzFlux (www.ozflux.org.au/) and Fluxnet (AU-
How) site (Beringer et al., 2016a) near Darwin in the Northern Territory, Australia. A record of
carbon, water and energy flux, as well as meteorological and soil measurements, was first established
at Howard Springs in 1997 (Eamus et al., 2001). As such, many detailed site descriptions exist
(Beringer et al., 2007;Hutley et al., 2013;Beringer et al., 2015;Moore et al., 2016a) so only a brief
description is provided here. Annual rainfall for the Howard Springs area is 1732 mm (± 44 SE) mm,
(Australian Bureau of Meteorology (BoM), station ID: 014015, www.bom.gov.au) of which 90-95 %
falls within the rainy (wet) season months of October to April. For this study, we defined the wet
season as a 6 month period from October 15[th] through to April 15[th] and the dry season as April 16[th] to
October 14[th] based on the work of Cook and Heerdegen (2001). Mean daily maximum air temperature
varies annually between 30.6 to 33.3 °C and mean daily minimum air temperature ranges from 19.3 to



25.3 ℃ (Australian Bureau of Meteorology, www.bom.gov.au/). Howard Springs is defined as a
mesic savanna because it receives >1200 mm rainfall annually (Hutley et al., 2011) and is classified
as 'open forest savanna' based on its canopy cover fraction (50-60 %) after Specht (1972). Soils are
mostly red Kandosols (Isbell, 1996) that are sandy-loamy, well weathered and nutrient poor.
Vegetation consists of a $C_3$ woody overstory that is dominated by evergreen *Eucalyptus miniata*
(Darwin woollybutt) and *E. tetrodonta* (Darwin stringybark). A smaller portion of the tree canopy and
mid-canopy layer is made up of semi-, brevi- and fully deciduous species such as *Erythrophleum*
*chlorostachys* (Ironwood) and *Terminalia ferdinandiana* (Kakadu plum) (Williams et al., 1997;Hutley
et al., 2011). Mean canopy height is 18 m (Hutley et al., 2011). The understory is dominated by the
annual $C_4$ grass *Sorghum intrans* (spear grass) and perennial $C_4$ grasses *Heteropogon triticeous* and *S.*
*plumosum*. Also sharing the understory with the grasses are saplings (juveniles) of overstory species,
the shrub *Buchanania obovata* and the cycad *Cycas armstrongii*. Due to the frequent occurrence of
fire in Australian savanna (Beringer et al., 2015), control burning was performed at the beginning of
each dry season to protect the monitoring equipment at the Howard Springs flux site.

## 2.2 Productivity measurements

To estimate productivity from the savanna ecosystem and partition it into tree (overstory) and grass
(understory) GPP, we used the eddy covariance technique (Baldocchi et al., 2001) as detailed for
Howard Springs by Moore et al. (2016a). Two eddy covariance towers were in operation at Howard
Springs to measure the fluxes of carbon, water and energy from both the understory (within tree
canopy tower at 5 m) and the ecosystem (above tree canopy tower at 23 m). The overstory flux
component is simply the difference between ecosystem and understory fluxes and represents the
above ground tree fluxes. Instrumentation, validation of the understory tower, data quality assurance
and quality control (QA/QC) and flux partitioning information is also provided in Moore et al.
(2016a). Therefore, we provide only a brief description of the site instrumentation and flux data
processing.
Core eddy covariance instruments on each tower consisted of a 3D sonic anemometer (CSAT3,
Campbell Scientific, Logan UT) and an infra-red gas analyser (LI-7500, Li-COR Biosciences, Lincoln,
NE). These instruments sampled at a rate of 10 Hz and provided 30-min flux averages. Soil heat flux
(HFT3, Campbell Scientific, Logan, UT) and net/short/long wave radiation components were also
recorded on the ecosystem tower (CNR4, Kipp and Zonen, Delft, NL). The raw 30-minute data were
QA/QC'd to level 3 standard using the OzFluxQC (v2.9.4) python scripts. Energy balance closure
analysis of the ecosystem tower, based on daily data (Leuning et al., 2012), gave a slope of 0.89 and
an $r^2$ of 0.92. The understory tower was validated via power spectra analysis (Moore et al., 2016a) that
followed idealised curves for vegetated canopies (Kaimal and Finnigan, 1994). Level 3 data were then
gap filled and used to partition net ecosystem exchange (NEE) into respiration and GPP using the



Dynamic INtegrated Gap filling and partitioning for OzFlux (DINGO, (Beringer et al., 2016b))
package. Both OzFlux and DINGO packages were written using python scripts.
**2.3 Phenology and light use efficiency (LUE) measurements**
Alongside the flux tower estimates of tree and grass productivity, we recorded time series of incident,
reflected and absorbed PAR, as well as vegetation cover change. While the understory is largely
homogenous in species distribution at the flux tower footprint scale (i.e. >50 m), variation from one
point to the next does exist in the understory due to its vegetation composition. To obtain a rigorous
time series, spatial replicate measurements of vegetation cover change and PAR variability are
required. We used five towers (mini towers), each of which were 5 m tall and made from steel square
hollow section with a cross arm to attach the instruments and a logger-solar panel array. The towers
were stabilised using guy wires and a base plate. A winch system was used to manoeuvre the
instruments up and down the tower for data download and maintenance. The towers were set up at a
distance of 50 m in a pentagon shape around the main ecosystem flux tower and faced an east-west
direction (Fig. 1).
To measure the components of PAR in the savanna, we installed PAR sensors on each of the mini
towers (SQ-Series, Apogee, Logan, UT). Incoming PAR reaching the understory, through the
overstory vegetation, was measured with a PAR sensor installed facing upward at 5 m on each mini
tower. Another sensor was installed facing downward at 5 m to record the amount of PAR reflected
by the understory vegetation. The amount of PAR reaching the ground surface through the understory
vegetation was recorded with a third PAR sensor facing upward at 10 cm. A data logger (CR800,
Campbell Scientific, Logan UT) and multiplexor (AM25T, Campbell Scientific, Logan, UT) were
used to collect and store PAR data and to operate the phenocams. The mini tower system was
powered using a 20 W solar panel, 12 V regulator and 12 V gel cell battery. To provide a complete
accounting of PAR in the savanna, two additional PAR sensors (LI-190 Quantum Series, Li-COR
Biosciences, Lincoln, NE) were installed on the 23 m flux tower for collection of incoming PAR and
outgoing PAR reflected from the savanna ecosystem.
Changes in savanna overstory and understory vegetation greenness were assessed using consumer-
grade point-and-shoot cameras (Canon Powershot A810). Each mini tower supported two cameras,
one to collect upward facing images of the tree canopy and one to collect downward facing images of
the understory, making a total of 10 cameras installed. The cameras were set to run using automatic
exposure in aperture priority mode, with a low f/stop value of 2.8 to ensure the entire image was used
to respond to ambient light levels (Richardson et al., 2007;Ryu et al., 2012;Sonnentag et al., 2012).
Automatic white balance was also used as we did not have a grey reference panel to correct for white
balance manually. Images were stored on SD memory cards in a compressed JPEG file format, to
ensure the cards did not fill between site visits.



Each camera was housed in a waterproof case with an aperture hole cut in the top that was sealed with
a microscope slide (Fig. 2, a & d). Following the concept of Ryu et al. (2012), power was delivered to
the cameras through wires soldered to the battery terminals, which received input from a 12 V relay
connected with a 3.3 V regulator. A second 5 V relay was used to send a short pulse to wires soldered
onto the 'on-button' of the cameras to mimic the action of turning the cameras on. The Canon Hack
Development Kit (http://chdk.wikia.com/wiki/CHDK) was used to modify the cameras to
automatically take an image when turned on, which was administered via a u-Basic script saved on
the memory card. Each mini tower logger was programmed to operate the cameras twice daily, once
at 11:30 ACST (to match the MODIS Terra overpass) and once at 13:00 ACST (approximately solar
noon). Each camera was installed on the mini towers using a metal plate angled at 57.5 ° from zenith,
as this angle has been found to minimise the effects of leaf inclination angle when calculating LAI
(Weiss et al., 2004;Baret et al., 2010).
**2.4 Phenocam image and radiation data processing**
Phenocam images were firstly visually checked for field of view (FOV) shifts and major obstructions
(i.e. water on the case windows) as a first step in the image QA/QC process. Images with obstructions
were removed, which accounted for between 3 - 13 % of images for each camera. However, three out
of ten cameras were completely omitted from analysis due to severe FOV shifts or where an
individual camera had greater than 50% of images lost. This left a total of four cameras for understory
analysis (5031 images total) and three for the overstory (4255 images total). All remaining cameras (n
= 7) experienced slight FOV shifts as a result of manual data download. However, a Student t-test of
686 analysed images, for a camera with a large visible FOV shift, revealed no significant effect on the
extracted results ($t_{686}$ = 0.13, p = 0.90). The time series from each camera were then gap filled using
the best regression relationship against another camera, most of which had an $r^2$ >0.8.
For each camera, the images were analysed using code written in python that initially took the
extracted Exif (Exchangeable image file format) data to rename files using a standardised (yyyy-mm-
dd hh:mm:ss) format. Images were analysed in date/time succession using a region of interest (ROI)
that encompassed as much of the vegetation as feasible. As a result, the ROI varied depending on the
vegetation available in the overstory FOV and was the same for all understory cameras, except for a
separate analysis of grass and woody green vegetation, which required individual ROI's for each
understory camera (Fig. 2).
Each camera collected 8-bit depth red-green-blue (RGB) colour channel information, stored as digital
numbers (DN), at a resolution of 4608 x 3456 pixels. These DN's provide a measure of colour
intensity based on irradiance, so they can be highly variable when scene illumination changes (Ide and
Oguma, 2010;Sonnentag et al., 2012). To reduce the effects of scene illumination, the DN's are



typically used to calculate the green (GCC) chromatic coordinate, a normalised ratio of the green
channel to all channels, as Eq. (1) (Gillespie et al., 1987;Woebbecke et al., 1995):
$Gcc = G_{DN}/(R_{DN} + G_{DN} + B_{DN})$  (1)
where DN is the digital number that corresponds with the green (G), red (R) and blue (B) channels.
The red (RCC) and blue (BCC) chromatic coordinates were calculated in the same way as GCC.
Chromatic coordinate values were calculated for each pixel within the ROI and then averaged to give
an overall GCC, RCC and BCC value for each image. In addition to the chromatic coordinates, we
also calculated the excess green (ExG), red (ExR) and blue (ExB) indices in order to compare which
colour index performed best in capturing savanna phenological change. The excess index is an
enhancement of the respective colour channel information against the other channels and is calculated
as Eq. (2) (Woebbecke et al., 1995):
$ExG = 2G_{DN} - (R_{DN} + B_{DN})$  (2)
The amount of light used by vegetation over time is directly correlated with productivity (Monteith,
1972). Using the PAR data collected from the mini towers, we calculated fPAR for the overstory (OS)
Eq. (3), understory (US) Eq. (4) and ecosystem (ECO) Eq. (5) as:
$fPAR_{OS} = (PAR_{AED} - PAR_{AGD})/PAR_{AED}$  (3)
$fPAR_{US} = (PAR_{AGD} - PAR_{BGD})/PAR_{AGD}$  (4)
$fPAR_{ECO} = (PAR_{AED} - PAR_{BGD})/PAR_{AED}$  (5)
where AED is the above ecosystem downwelling PAR, AGD is the above grass downwelling PAR,
and BGD is the below grass downwelling PAR. We did not include the reflected component of PAR
in our calculations as this consistently produced negative fPAR results in the dry season. Once fPAR
was calculated, APAR was calculated for overstory, understory and ecosystem by multiplying the
respective fPAR with available incoming PAR (note: this was $PAR_{AGD}$ for the understory).
The variability over time of vegetation LAI and biomass is a direct result of phenology and
productivity. We collected overstory LAI on each site visit (6 total) and understory biomass samples
spanning a full growing season (Dec-Apr, 4 total) to investigate how these variables changed
alongside the flux and phenocam data. We used digital hemispheric photography to record overstory
LAI using a Canon digital single lens reflex (DSLR) camera (Rebel T1i) with a 185 ° super fisheye
FOV (f/5.6) lens attached. A one hectare plot was established around the central ecosystem flux tower
and LAI measurements were recorded every 20 m within it (n = 36, Fig. 1). These images were
analysed using WinScanopy (v2014a), where a clumping coefficient was calculated to account for
foliage clumping in the LAI estimate.



A Tracing Radiation and Architecture of Canopies (TRAC) instrument was used to verify the
WinScanopy clumping index parameter, which agreed within 10-15 % of each other (0.82 to 0.94 in
the wet season, 0.61 to 0.67 in the dry season) and gave us confidence in the hemispheric LAI
estimates. Understory biomass below 2 m in height was collected from 20 replicate 1 x 1 m quadrats
along a N-S and E-W 100 m transect (10 samples each, every 5 m) and separated in the lab into grass
and other green biomass, weighed, then oven dried at 80 °C for 3 days to obtain a dry weight. The
exact distance of sampling along each transect was altered by 1 m for each site visit to avoid biasing
from the previous sampling period. Following the technique used by Chen et al. (2003), we converted
the dry weight biomass into carbon content assuming it to be 43 % of grass biomass and 49 % of other
green biomass.
**2.5 Light use efficiency (LUE) models and incorporation of phenology**
An alternative to estimating GPP from flux towers is to use a LUE model, where GPP is
approximated by relating plant productivity to the amount of light they absorb over a growing season
(Monteith, 1972). The MODIS GPP product (MOD17 A2/A3) is calculated using a LUE model (Eq.
(6), Running and Zhao, 2015), which we use in this study, as it has been previously validated for
Australian savannas (Kanniah et al., 2009):
$$GPP = APAR \times LUE_p \times T_{MIN}scalar \times VPDscalar \qquad (6)$$
where GPP is in g C m$^{-2}$ d$^{-1}$, APAR is in MJ and LUE$_p$ is peak light use efficiency in g C MJ$^{-1}$ PAR$^{-1}$.
Because C$_3$ (trees) and C$_4$ (grasses) plants have different maximum LUE rates (Zhu et al., 2008), we
calculated overstory and understory LUE$_p$ separately following a similar approach to Kanniah et al.
(2009) and Coops et al. (2007), where LUE is firstly calculated as GPP/APAR and is then binned by
month to obtain monthly LUE. We chose to use the months of Dec-Mar (inclusive) to provide an
estimate of LUE$_p$ for the overstory and understory, as these months ($n = 8$) have the least
environmental constraints to productivity and should be close to the maximum. This gave us a LUE$_p$
value of 1.49 ± 0.06 g C MJ$^{-1}$ PAR$^{-1}$ for the ecosystem, 1.22 ± 0.03 g C MJ$^{-1}$ PAR$^{-1}$ for the overstory
and 2.41 ± 0.23 g C MJ$^{-1}$ PAR$^{-1}$ for the understory (Fig. 3). In the LUE model the LUE$_p$ values are
then down regulated on a daily basis using the VPDscalar Eq. (7) and T$_{MIN}$scalar (values between 0
and 1 ) Eq. (8) (Running and Zhao, 2015):
$$VPDscalar = (VPD_{max} - VPD_d)/(VPD_{max} - VPD_{min}) \qquad (7)$$
$$T_{MIN}scalar = (T_{MIN} - T_{MIN\,min})/(T_{MIN\,max} - T_{MIN\,min}) \qquad (8)$$
where T$_{MIN}$ is the minimum daily temperature for a given day, T$_{MINmax}$ is the minimum daily
temperature when LUE is at maximum and T$_{MINmin}$ is the minimum daily temperature when LUE is 0,
all of which are output in °C. Likewise, VPD$_d$ is the mean daytime VPD, VPD$_{max}$ is the maximum



VPD when LUE is 0, and $VPD_{min}$ is the minimum VPD when LUE is at maximum, all output in Pa.
These scalar values fall between the range of 0 – 1. The MOD17 GPP algorithm uses values of -8 °C
for $T_{MINmin}$, 11.39 °C for $T_{MINmax}$, 650 Pa for $VPD_{min}$ and 3500 Pa for $VPD_{max}$ for savannas (Running
and Zhao, 2015), so we also used these values for Howard Springs.
The use of a soil moisture term, evaporative fraction (EF), has been argued to represent plant
available moisture more reliably than VPD (Gentine et al., 2007;Yuan et al., 2007;Kanniah et al.,
2009). This term is simply a fractional estimate of latent heat (LE) divided by the sum of sensible heat
(H) and LE (i.e. LE/ (LE + H)). We also used the EF term in this study to test if and how it improved
the estimation of overstory, understory and ecosystem GPP. For the overstory and ecosystem, we
calculated EF using the ecosystem flux tower, whereas for the understory we calculated EF using the
understory flux tower.
Another technique we tested for improving GPP estimates from the LUE model was to input
phenocam greenness indices, as they have been found to correlate with ecosystem productivity in
northern hemisphere forests and grasslands (Richardson et al., 2009b;Migliavacca et al.,
2011;Toomey et al., 2015). We hypothesised that inclusion of GCC in the LUE model would improve
the model's ability to predict savanna overstory and understory GPP, particularly given the strong
phenology cycles displayed in savannas. As GCC is a fractional measure, like that of fPAR, we
substituted GCC as a proxy for fPAR using the coefficients of a regression to normalise it, a similar
approach to that used by Migliavacca et al. (2011). As a result, Eq. 6 was transformed to include
$PAR\cdot(mGCC+c)$ in place of APAR, where $m$ and $c$ are the linear regression coefficients.
We repeated the above technique using MODIS EVI (Huete et al., 2002), to test if satellite indices
could be used to improve estimates of ecosystem scale GPP. We chose the EVI product
(MOD13Q1.005) as it has been shown to function well for identifying broad-scale phenology in
Australian savannas (Ma et al., 2013;Ma et al., 2014). A 3 x 3 pixel cut out of EVI data surrounding
the Howard Springs site, at 16-day and 250 m resolution, was processed in DINGO accepting the
quality flags 00 (highest overall quality) and 01 (good quality) only. The 16-day data were then
interpolated and smoothed, using a Savitzky-Golay technique (Savitzky and Golay, 1964) in DINGO,
to create a daily time series of EVI (Beringer et al., 2016b). Daily EVI were regressed against site-
based daily ecosystem fPAR and the regression was used to replace APAR in Eq. (6) to estimate
ecosystem GPP.
Finally, to test the performance of each model against tower GPP estimates, we used a Pearson
correlation to provide a closeness of fit estimate (Corr) and test if the relationship was statistically
significant ($p<0.05$). We also calculated the root mean squared error (RMSE) to provide a measure of
the difference between the two datasets (tower and model) and the relative predictive error (RPE) to





represent the percentage difference between them, plus the degree of over- (+) or underestimation (-)
of the model.
**3 Results & Discussion**
**3.1 Phenological insights from phenocams**
Extraction of the chromatic coordinates and excess indexes revealed expected patterns from overstory
and understory vegetation over time, showing that the cameras functioned well as phenology monitors
of vegetation greenness at the ecosystem and individual species level (Fig. 4, 5 & 6). Not surprisingly,
both GCC and ExG were at their highest in the understory during the wet season and gradually
declined to their lowest values by the late dry season (i.e. September, Fig. 4). The RCC/ExR indices
showed an inverse relationship to GCC/ExG, which is usually symptomatic of increased red
pigmentation due to senescing leaves and chlorophyll loss (Hoch et al., 2001;Lee et al., 2003;Keenan
et al., 2014). This relationship is shown by the red-green index crossover in the understory that
coincides with grass senescence and signals the end of the wet season (i.e. Mar/Apr, see Fig. 4).
However, in this case, the crossover is likely the combined result of grass senescence (loss of green)
and an increase in the red kandosol soil background showing through with the loss of understory
biomass.
A different story is depicted at the beginning of the wet season (Oct/Nov), whereby the red-green
crossover does not occur as quickly as it does at the end of the wet season (Fig. 4). Here it occurs after
several rainfall episodes in November (Fig. 4, Fig. 5 for rainfall). This is due to the time needed for
the vegetation, particularly the grasses, to respond to the onset of the rainy season, as October and
November are typically build up months where convective storms deliver rain in single events before
the onset of the more consistent monsoonal rain in December (Cook and Heerdegen, 2001). Peak
GCC and ExG are not reached until February (Fig. 4), which is also reflected in results from the
biomass harvest (Table 1) that shows the mid wet season (February) to be the period of highest
productivity for total understory biomass.
The understory consists of a mix of annual (*S. intrans*) and perennial (*S. plumosum* & *H. triticeous*)
grasses, saplings of overstory (*E. tetrodonta* & *E. miniata*) and mid-story (*E. chlorosyachys*, *T.*
*ferdinandiana* & *B. obovata*) species and cycads (*C. armstrongii*) that all have differing phenologies
(Bowman and Prior, 2005). The behaviour of these phenological guilds is reflected in the temporal
patterns of GCC and ExG between grasses and the non-grass woody elements (herein referred to as
'woody green') and provides additional insight into the dynamic nature of understory savanna
phenology (Fig. 5). While grasses are considered the most abundant of the understory species in terms
of biomass (Table 1) and LAI at Howard Springs (Hutley et al., 2000), they are only active during the
wet season months (Andrew and Mott, 1983;Scott et al., 2010). During both the early dry season





(April/May) and after the first rains of the wet season (October/November), the woody green species
take advantage of the lack of grass to gain biomass (Werner and Franklin, 2010;Werner and Prior,

3   2013).

Annual grasses typically germinate after the first 15 mm or more of rainfall, with further rainfall
events required to drive leaf growth (Andrew and Mott, 1983;Cook et al., 2002). Pre-monsoonal
rainfall is highly variable in terms of its timing and amount, therefore this phenological strategy may
minimise the possibility of grass mortality if dry periods proceed an initial early wet season rainfall
event (Mott, 1978). In Fig. 5, this delay in grass greening is evident, with rapid increases in GCC only
occurring approximately a month after the first rainfall event (Fig. 5, Oct-Dec 2013). Such detailed
analysis of the phenocam data can tease apart composite greening signals to better understand
phenological dynamics and fluxes (Fig. 5 & 7). Results from understory biomass harvests also support
the GCC results, revealing that as the wet season progressed, grass biomass increased in dominance to
account for 77 % of understory biomass by the end of the wet season (Table 1). This shows that while
the grasses are the primary driver of understory biomass and productivity, the woody green species
also make important contributions throughout the year and are likely the reason why understory GPP
does not completely cease in the dry season (Moore et al., 2016a).
In contrast to the understory, the overstory GCC and ExG did not fluctuate much in comparison to
their red and blue channel indices (Fig. 6, a). This is mostly due to the high portion of blue sky and
cloud within the ROI's for the overstory images (Fig. 2, e & f), which vary depending on daily
weather conditions. The effect on BCC/ExB is particularly strong during the wet season, where the
summer monsoon varies sky conditions considerably between bright blue sky and dull grey cloud.
Due to the narrow FOV and upward orientation of the overstory cameras, the trees were prone to
moving in and out of smaller ROI tested (data not shown), so a larger ROI was chosen to ensure tree
foliage was always present in the image, but at the cost of including a greater sky portion (Fig. 2, e &
f).
Nevertheless, when viewed in isolation from the red and blue indices, temporal variation was apparent
in overstory GCC and ExG. These indices captured the variability in greenness and were consistent
with changes in overstory LAI when compared with adhoc hemispherical LAI measurements (Fig. 6,
b). While there is inherent uncertainty in both the phenocam imagery (i.e. FOV, scene illumination)
and LAI (i.e. leaf projection and orientation, clumping, gaps, see Ryu et al. (2010)) estimates in this
study, the savanna tree canopy is known to experience seasonal fluctuations in LAI with the highest
values in the wet season and lowest values in the late dry season (Williams et al., 1997;O'Grady et al.,
2000). The same general pattern is displayed in Fig. 6, giving us confidence that the phenocams are
able to detect overstory cover change, despite their limitations in setup and image collection.



### 3.2 Integrating phenocam and MODIS phenology with GPP

The seasonality of GPP in these savannas has been found to differ between that of the overstory and understory, with understory GPP tied more closely to the duration of the wet season than that of the overstory (Moore et al., 2016a). The GCC and ExG time series approximated the overstory and understory GPP estimates well (Fig. 7), and we hypothesise that they could be useful for independently predicting overstory and understory GPP. Simple linear regressions of GCC against flux tower GPP quantified the relationship between the two variables, with understory GPP ($r^2 = 0.65$) revealing a closer fit with GCC than overstory GPP ($r^2 = 0.23$, Fig. 7). The ExG index did not perform so well compared with GCC for the overstory ($r^2 = 0.09$) but improved the relationship slightly against GCC for the understory ($r^2 = 0.70$, Fig. 7). ExG was originally developed for identifying green vegetation from images with a soil background (Woebbecke et al., 1995). This is a likely reason for why the relationship between ExG and GPP was slightly closer to 1:1 than that of GCC for the understory.

While the relationship between overstory greenness (ExG/GCC) and GPP is not as strong as that of the understory, the phenocams are still able to detect seasonality in greenness that follows GPP over time (Fig. 7). The trees have a deeper rooting structure than the grasses, allowing them to access a larger volume of soil moisture (Eamus et al., 2002;Kelley et al., 2007) and thus maintain constant overstory transpiration throughout the year (O'Grady et al., 1999;Hutley et al., 2000). While the tree canopy is largely evergreen, the LAI will drop up to 30-40 % in order to account for the dry season water deficit (O'Grady et al., 2000;Whitley et al., 2011), which is also apparent from both our overstory LAI and GCC results (Fig. 6). Tree productivity, in contrast to transpiration, is known to decrease into the dry season (Eamus et al., 1999), and most carbon uptake is directed toward maintenance respiration rather than growth (Chen et al., 2002;Prior et al., 2004;Cernusak et al., 2006). However, the occurrence of late wet season rainfall events may benefit the productive capacity of the trees by boosting soil moisture stores, thereby supporting higher rates of productivity for longer in the dry season (Moore et al., 2016a). This effect is apparent in our overstory GCC time series, where after late April to early May rainfall events (see Fig. 5 for daily rainfall), GCC spikes in June indicate a flushing of the foliage in the dry season (Fig. 6, b).

At the ecosystem scale, the interaction of the overstory and understory with the wet and dry seasons drives variability in productivity. The MODIS greenness index, EVI, mostly captures this variability, albeit at coarser temporal resolution (Fig. 7, e) when compared with the phenocams. While the broad scale variability in savanna phenology change is captured by EVI, such as seasonality (Ma et al., 2013), it is not able to capture the finer scale details that the site based phenocams can. MODIS indices, such as EVI, do not currently have the ability to identify individual plant scale phenology patterns (Brown et al., 2016;Moore et al., 2016b), which is another advantage of the phenocam (Fig. 5). The phenocam data also provides a useful means of validating the MODIS data in that both are



able to track the seasonality of savanna GPP, which is driven by a complex interaction of both
meteorology and phenology   (Kanniah et al., 2011;Whitley et al., 2011;Ma et al., 2013;Ma et al.,

3    2014).

**3.3 Integrating phenocam and MODIS phenology with a LUE model**
In order to test the applicability of the phenocam indices and MODIS EVI to independently predict
savanna GPP using a LUE model, peak LUE (i.e. $LUE_p$) needed to be calculated for the ecosystem,
overstory and understory. The calculated $LUE_p$ value was higher for the understory ($2.41 \pm 0.23$ g C
$MJ^{-1} PAR^{-1}$) compared to the overstory ($1.22 \pm 0.03$ g C $MJ^{-1} PAR^{-1}$, Fig. 3). The higher $LUE_p$ for the
understory is largely due to the dominance of $C_4$ grasses in the understory (Table 1), as their $C_4$
photosynthetic pathway is more energy efficient (Sage, 2004;Osborne and Beerling, 2006;Zhu et al.,
2008). Our values fell within the range of $LUE_p$ reported for African savannas, which have varied
from as low as 0.33 g C $MJ^{-1} PAR^{-1}$ up to 3.5 g C $MJ^{-1} PAR^{-1}$ depending on the vegetation and season
(Sjöström et al., 2013;Tagesson et al., 2015). Recent work has shown the importance of correctly
applying $LUE_p$ values to $C_3$ and $C_4$ plants when using LUE models to calculate GPP (Yan et al., 2015).
Therefore, to account for the $C_3$:$C_4$ differences, we applied these site and trait specific values to the
LUE model used to estimate GPP.
The next step in our parameterisation of the LUE model was to test it in its traditional form; using the
meteorological inputs of $T_{MIN}$ and VPD that constrain $LUE_p$, along with APAR (Eq. 6). We found the
model captured most of the seasonality of overstory GPP but underestimated the magnitude of GPP in
the dry season and overestimated GPP in the wet season (Table 2, Fig. 8, a). For the understory, the
LUE model appeared to overestimate and lag flux tower GPP consistently by 1-2 months (Table 2,
Fig. 9, a). This resulted in a strong dry season over estimate of understory GPP (159 %, Table 2). For
the ecosystem, the LUE model consistently overestimated GPP (Table 2, Fig. 10, a). Kanniah et al.
(2009) also found the LUE model performed poorly for the Howard Springs ecosystem, so they
replaced the standard VPD parameterisation with an EF term and found this to improve the
relationship, which we implemented next.
Application of EF to the overstory model in this study improved its ability to predict GPP in the dry
season but overestimated GPP in the wet season, causing an over prediction of annual GPP by 8 %
overall (Table 2, Fig. 8, a vs. b). In contrast, the inclusion of EF in the understory LUE model slightly
improved the prediction of annual GPP, with better correlation (0.73 vs 0.57), lower RMSE (1.43 vs.
2.02 g C $m^{-2}$) and lower RPE (22.27 vs. 62.09 %). However, the understory model still lagged tower
GPP and was still particularly poor at capturing the seasonal transitions (Fig. 9 a & b). For the
ecosystem, the inclusion of EF enhanced the overestimation of GPP from 14 to 25 %, particularly in
the wet season (Table 2, Fig. 10, a vs. b). EF provides a proxy measure of soil moisture as it includes
a water flux component (LE) that is tightly linked with soil moisture availability (Gentine et al.,





2007;Kanniah et al., 2009). In Australian savannas, soil moisture is highly seasonal and a major driver
of productivity (Kanniah et al., 2010). This makes EF a useful index in the dry season, when latent
heat largely comes from transpiration and is therefore tightly coupled with GPP. However, in the wet
season, soil evaporation contributes a large amount to latent heat, which is not tightly coupled to GPP
(Kanniah et al., 2009). This explains why EF is able to constrain the LUE model in the dry season and
why it performs poorly in the wet season and transition periods.
The incorporation of phenocam GCC into the LUE model improved the estimate of understory GPP
substantially (Table 2, Fig. 9, c & d). This was most apparent with the combined use of GCC and EF
in the LUE model, which produced the best correlation (r = 0.85), lowest RMSE (0.96 g C m$^{-2}$) and
lowest RPE (17.73 %, Table 2, Fig. 9). These results show that while EF is an important factor for
GPP, greenness phenology is also key for estimating understory productivity. In further support of
this, the inclusion of GCC also eliminated the lag in model estimated GPP, bringing the estimate
closer in line with seasonal variability from the flux tower, as evidenced by the large decrease in
RMSE and RPE (Table 2, Fig. 9). As previously discussed, the understory grasses (annual species in
particular) die off at the cessation of the wet season and do not contribute to the small fraction of
understory GPP in the dry season (Moore et al., 2016a). This is a plant phenology response, rather
than a response to meteorological conditions, as factors such as soil moisture remain high enough in
the early dry season to support plant growth (Eamus et al., 2002;Kelley et al., 2007;Moore et al.,
2016a). Given that these grasses dominate understory biomass at Howard Springs, it is not surprising
that including greenness phenology information in the LUE model improves its output relative to the
flux tower.
The inclusion of greenness indices in the LUE model for the overstory (GCC) and ecosystem (EVI)
also improved the estimate of GPP. For the overstory, the combination of EF and GCC performed
slightly better in the dry season than GCC alone, but was not able to capture the wet season well
(Table 2, Fig. 8 d). This resulted in the incorporation of GCC into the LUE model producing the best
overall result, despite the slightly lower correlation value (0.60 vs 0.72) and RMSE (1.43 vs. 1.36 g C
m$^{-2}$) when compared with GCC and EF combined (Table 2). For the ecosystem, the inclusion of EVI
into the LUE model performed the best at predicting GPP, which was supported by the lowest values
for RMSE (2.03 g C m$^{-2}$) and RPE (13.76 %, Table2).
The greenness information clearly fills an important gap in relation to changes in overstory,
understory and ecosystem greenness. The general improvement in LUE model output for overstory,
understory and ecosystem with the inclusion of greenness phenology information highlights the
importance of accounting for phenological variability when estimating GPP in savannas. A similar
result was found for a subalpine grassland in Italy, where phenocam greenness indices improved the
ability of the same LUE model to predict grassland GPP (Migliavacca et al., 2011). Likewise, in an



evergreen Amazonian rainforest, Wu et al. (2016) linked phenological changes in leaf development
and demography to seasonality in GPP, showing the importance of phenology as a driver of
ecosystem productivity. For Australian savannas, the effect of phenology is most evident at the end of
the wet season (Apr-May), where in the understory, growth ceases due to annual grass senescence
even though meteorological conditions (temperature, VPD and/or EF) are still sufficient to support
growth (Fig. 9 a&b vs. c&d). The original LUE model over-predicts GPP as a result of this, which is
substantially reduced by the inclusion of greenness phenology indices.
**3.4 Limitations, impacts and further work**
While phenocams have consistently proven to be a useful tool for phenological and productivity
research (Richardson et al., 2009b;Migliavacca et al., 2011;Toomey et al., 2015;Wu et al., 2016),
there still remain several limitations that require further investigation to improve their utility. Issues
related to camera choice and image collection have been shown to be less problematic for simple
identification of phenological transition dates and seasonal variation than first thought (Sonnentag et
al., 2012), however, maintaining similar protocols for cross site comparisons remains preferable
(Moore et al., 2016b). Scene illumination variability is probably the most problematic limitation of
phenocams, which can be reduced by using chromatic coordinates or excess values, as well as by
setting the white balance to a fixed level (Richardson et al., 2009a;Ide and Oguma, 2010;Migliavacca
et al., 2011). Although white balance was not fixed for this study, we found that the GCC and ExG
time series matched well with GPP estimates regardless and provided added value to that gained from
using just APAR alone in the LUE model.
The wet season influence on scene illumination adds daily noise to the time series, but the indices are
still useful for informing seasonal productivity estimates. This same relationship will likely not stand
for other, less dynamic ecosystems in Australia (Restrepo-Coupe et al., 2015;Moore et al., 2016b), so
we recommend the fixing of white balance where appropriate. The use of a grey reference panel for
normalising phenocam images has also been proposed (Richardson et al., 2009a), however, this
technique has issues related to panel orientation and illumination conditions that can be different to
those experienced by the phenocams (Migliavacca et al., 2011). Despite these limitations, phenocams
are still an important tool for both species and plot scale phenology monitoring and with further
developments, will continue to provide valuable insight into Australian vegetation phenology (Moore
et al., 2016b).
In addition to the phenocam issues, the light use efficiency model used in this study is also subject to
limitations. This model relies on the input of meteorological information to generate an estimate of
ecosystem GPP. It is often found that these models overestimate GPP in the transition periods from
wet-dry or dry-wet in savanna ecosystems (Kanniah et al., 2009). The primary reason for this is that
savanna GPP is not driven solely by meteorology, that plant phenology also plays an important role,




as shown in our analysis. The technique for estimating $LUE_p$, used in the LUE model (Eq. 5), also
involves a degree of uncertainty that is centred around the input parameters of LUE and APAR, as
well as the scalars used to constrain it (Heinsch et al., 2006;Sjöström et al., 2013).
The MODIS MOD17 A2/A3 GPP product uses a $LUE_p$ value of 1.21 g C $MJ^{-1}$ for savannas and 1.24
g C $MJ^{-1}$ for woody savannas (Zhao and Running, 2010). While these values are close to the number
we calculated for the overstory (1.26 ± 0.03 g C $MJ^{-1}$ $PAR^{-1}$), we found the understory $LUE_p$ to be
much larger (2.44 ± 0.23 g C $MJ^{-1}$ $PAR^{-1}$). Similarly, for African savannas, $LUE_p$ has been found to
reach up to 3.50 g C $MJ^{-1}$ $PAR^{-1}$ in the wet (growing) season (Sjöström et al., 2013;Tagesson et al.,
2015). These $LUE_p$ values are much larger than that used in the MOD17 A2/A3 algorithm, which
suggests that tree-grass ($C_3$ vs. $C_4$) ratios need to be better accounted for in the LUE model. Recent
work from Yan et al. (2015) has shown this to be the case, where the application of different $LUE_p$
values to $C_3$ (1.8 g C $MJ^{-1}$ $PAR^{-1}$) and $C_4$ (2.76 g C $MJ^{-1}$ $PAR^{-1}$) plants improved global model
estimates of GPP.
Finally, the flux tower estimates of GPP are not without their own limitations, as the towers measure
NEE that is then partitioned into GPP and respiration most commonly by using a friction velocity (u*)
threshold at night and upscaling method for the daytime (Reichstein et al., 2005;Lasslop et al.,
2012;Barr et al., 2013). Use of the u* technique has been shown to be problematic at sites with
complex terrain (van Gorsel et al., 2009), where drainage flows result in horizontal loss of carbon
from an ecosystem that is not accounted for by the flux instruments. While Howard Springs is a
relatively flat site (slope < 1 º) that should prevent issues with using the u* technique, the flux tower
estimates from this site should still be considered with an amount of uncertainty as well (Moore et al.,
2016a;McHugh et al., In Submission). However, these issues have been addressed by previous work
at this site (Moore et al., 2016a) so we have confidence in the fluxes used for this study. Despite these
limitations, we were able to show that the input of phenological information into LUE models can
provide a useful constraint for estimating GPP within the uncertainty limits of tower derived estimates,
a similar conclusion to that found over a subalpine grassland in the Italian Alps (Migliavacca et al.,

27 2011).

**4 Conclusion**
We have shown the utility of phenocams for the monitoring of tree and grass phenology in savannas
and how this data can improve the quantification of productivity. Phenocams offer the ability to
decipher species level phenological signals, as shown by our time series analysis of understory grasses
and woody green species, as well as in the tracking of seasonal overstory leaf area change. Phenocams
have also shown to be useful for improving LUE models that have traditionally failed to capture the
wet-dry season transition periods well in savannas, which are characterised by phenology changes in





the understory that are out of sync with meteorological variability. This approach needs to be tested in
more ecosystems to determine its applicability for a wider range of ecosystem types, but promises
improved results for better understanding of ecosystem GPP and phenology. Phenological information
offers an important link for our understanding of ecosystem function as it provides a more accurate
means of independently verifying tower derived GPP estimates in savannas. We have demonstrated
that phenocams can be used in conjunction with eddy covariance flux towers to improve current
knowledge of savanna productivity and phenology, which will assist in our understanding of how the
tree-grass relationship in savannas may alter in the future.
**Author Contributions**
Field work and experimental design was executed by C. Moore, J. Beringer, L. Hutley and B. Evans.
Data analysis was chiefly carried out by C. Moore, with some coding assistance from B. Evans. The
manuscript was prepared by C. Moore with contributions from all co-authors.
**Acknowledgements**
Firstly, the authors would like to acknowledge support and funding from OzFlux and the overarching
Terrestrial Ecosystem Research Network (TERN), which is supported by the Australian Government
through the National Collaborative Research Infrastructure Strategy. This work utilised data collected
by grants funded by the Australian Research Council (DP0344744, DP0772981 and DP130101566).
Beringer is funded under an ARC FT (FT110100602). B. Evans is funded by the TERN Ecosystem
Modelling and Scaling Infrastructure. Special thanks are also made to Dr Peter Isaac for his
development of the OzFluxQC standardised processing tools and to Mr. Matthew Northwood for his
design and building of the mini towers and for his assistance with field work.





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



**Table 1:** Understory biomass harvest information for Howard Springs savanna collected across the wet seasons from 2012 to 2014.

| Period | Grass biomass (t ha$^{-1}$) | Other biomass (t ha$^{-1}$) | Grass biomass (%) | Other biomass (%) |
|---|---|---|---|---|
| Start Wet – Dec | 0.46 | 0.96 | 33 | 67 |
| Mid Wet – Feb | 1.34 | 1.77 | 43 | 57 |
| Peak Wet – Mar | 1.55 | 1.09 | 59 | 41 |
| End Wet - Apr | 1.31 | 0.38 | 77 | 23 |





**Table 2:** Summary of model performances against flux tower estimated GPP for overstory and understory at Howard Springs. Statistics include the Pearson Correlation coefficient (Corr), the root mean square error (RMSE, g C m⁻² d⁻¹) and the relative predictive error (RPE, %) for the light use efficiency model (LUE), LUE with evaporative fraction (LUE_EF), LUE with green chromatic coordinates (LUE_GCC) and LUE with EF and GCC (LUE_EF_GCC). The * highlights that the MODIS enhanced vegetation index (EVI) is used instead of GCC for the Ecosystem analysis. Pearson p values are not included as all were significant with P<0.001.

| | Model | Overstory | | | Understory | | | Ecosystem* | | |
|---|---|---|---|---|---|---|---|---|---|---|
| | | Corr | RMSE | RPE | Corr | RMSE | RPE | Corr | RMSE | RPE |
| All years | LUE | 0.64 | 1.45 | -1.79 | 0.57 | 2.02 | 62.09 | 0.80 | 2.11 | 13.77 |
| | LUE_EF | 0.73 | 1.47 | 8.40 | 0.73 | 1.43 | 22.27 | 0.79 | 2.69 | 25.18 |
| | LUE_GCC/EVI* | 0.60 | 1.43 | -0.85 | 0.78 | 1.40 | 55.32 | 0.81 | 2.09 | 14.85 |
| | LUE_EF_GCC/EVI* | 0.72 | 1.36 | 8.44 | 0.85 | 0.96 | 17.73 | 0.83 | 2.48 | 25.51 |
| Wet Season (15 Oct - 15 Apr) | LUE | 0.61 | 1.59 | 13.30 | 0.33 | 2.28 | 30.89 | 0.72 | 2.42 | 15.53 |
| | LUE_EF | 0.68 | 1.85 | 22.76 | 0.43 | 1.92 | 21.60 | 0.66 | 3.18 | 25.26 |
| | LUE_GCC/EVI* | 0.61 | 1.50 | 14.02 | 0.59 | 1.43 | 26.95 | 0.74 | 2.42 | 18.92 |
| | LUE_EF_GCC/EVI* | 0.70 | 1.67 | 23.13 | 0.66 | 1.25 | 19.33 | 0.71 | 3.07 | 28.36 |
| Dry Season (16 Apr – 14 Oct) | LUE | 0.37 | 1.32 | -17.97 | 0.54 | 1.76 | 159.34 | 0.56 | 1.80 | 11.21 |
| | LUE_EF | 0.63 | 1.05 | -7.76 | 0.48 | 0.84 | 24.36 | 0.71 | 2.19 | 25.06 |
| | LUE_GCC/EVI* | 0.24 | 1.37 | -16.80 | 0.41 | 1.38 | 143.73 | 0.39 | 1.76 | 8.94 |
| | LUE_EF_GCC/EVI* | 0.57 | 1.03 | -7.31 | 0.32 | 0.62 | 12.73 | 0.63 | 1.84 | 21.38 |





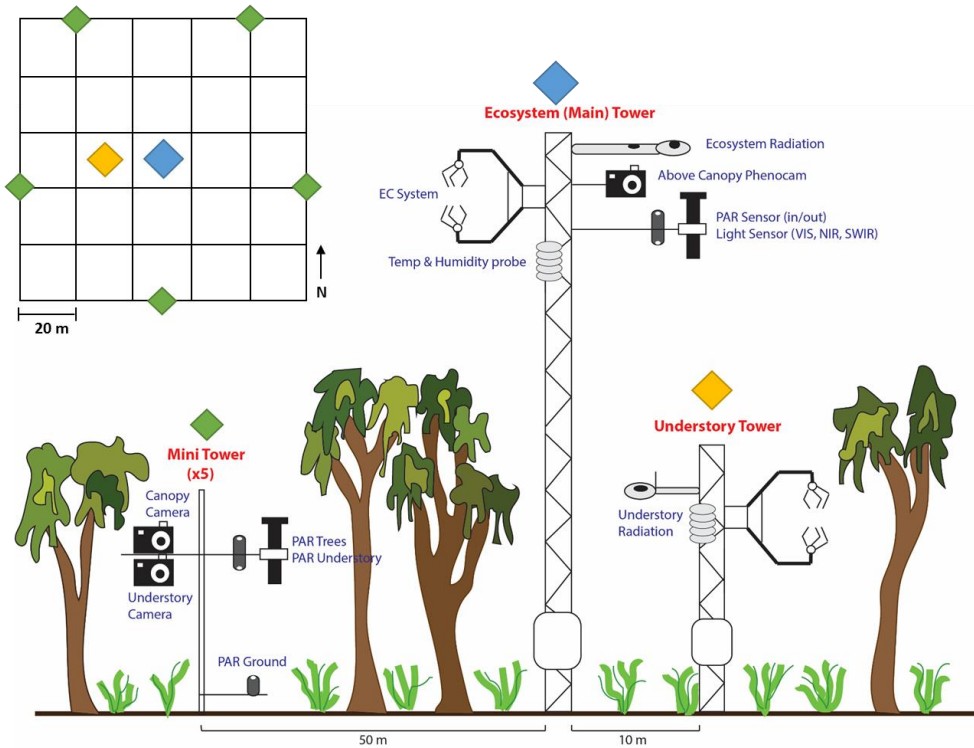

**Figure 1:** Diagram showing the core instrumentation supported by each flux tower and mini tower at the Howard Springs OzFlux site, as well as the layout of the monitoring plot.





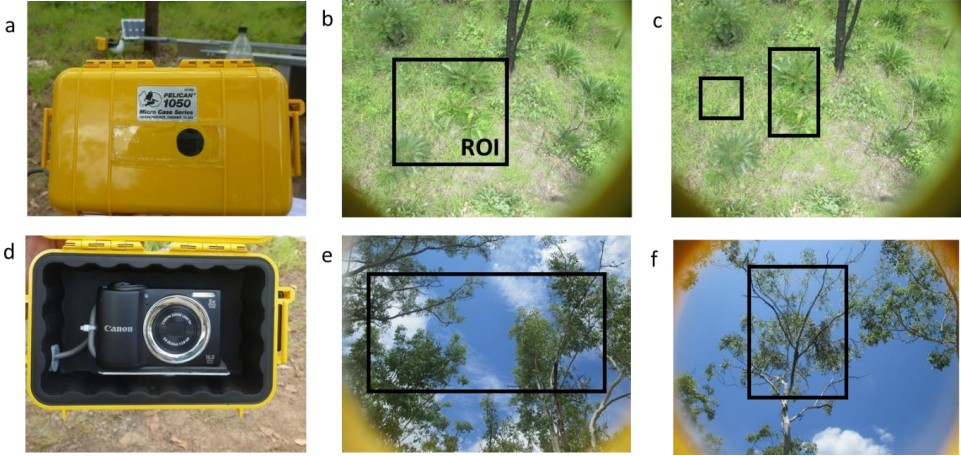

**Figure 2:** Camera setup (a & d) and examples of understory (b & c) and overstory (e & f) regions of interest (ROI, black box) used from phenocam images collected at the Howard Springs OzFlux site.



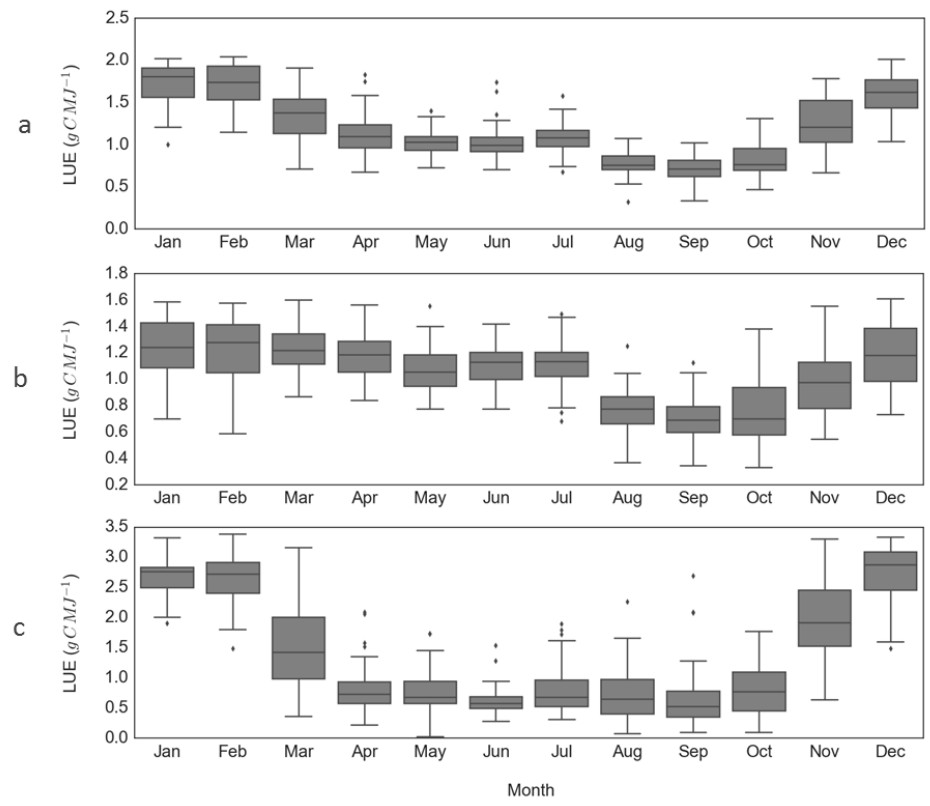

**Figure 3:** Monthly mean light use efficiency (LUE) ± SE (boxes) with 95 % confidence (whiskers) for the Howard Springs OzFlux site ecosystem (a), overstory (b) and understory (c) from December 2012 to October 2014. Individual dots represent outlier values for each respective month.





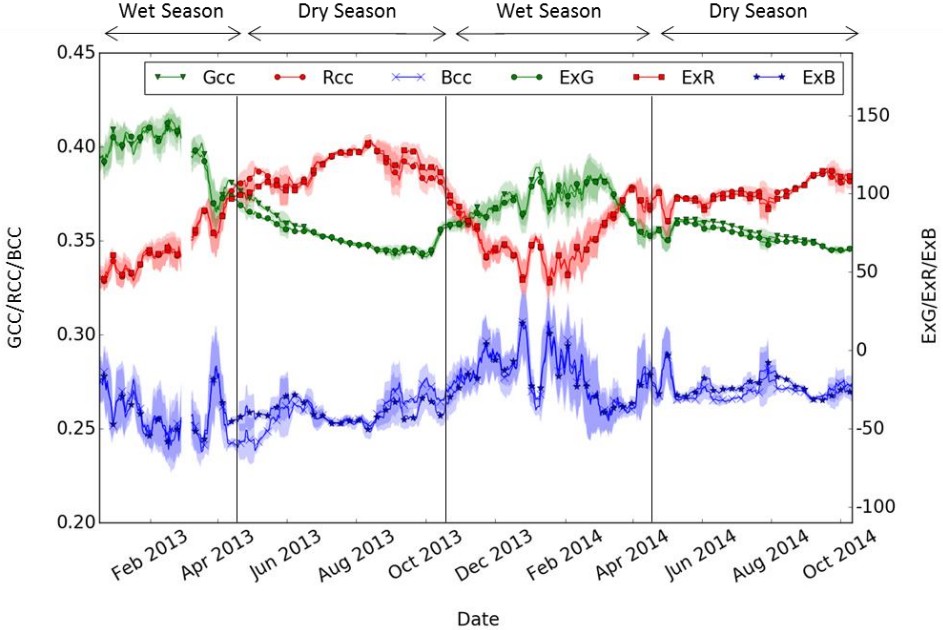

**Figure 4:** Daily green, red and blue chromatic coordinates (GCC/RCC/BCC) and excess indices (ExG/ExR/ExB) for the Howard Springs OzFlux site understory from December 2012 to October 2014. Daily data are shown with an 8-day centred running mean (marked every 8 days for visualisation) applied. The standard error of the mean is given by the shading.





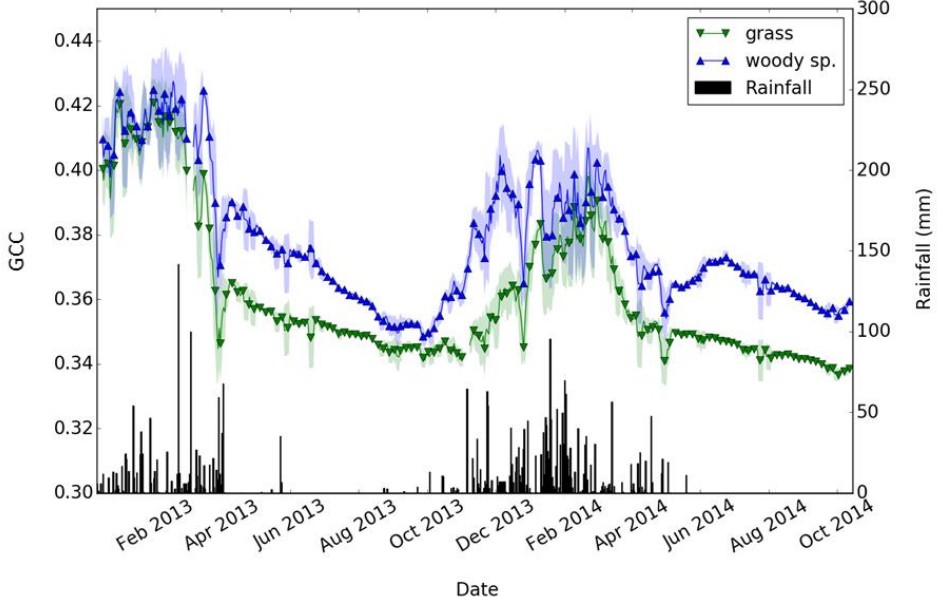

**Figure 5:** Daily rainfall (mm) and green chromatic coordinate (GCC) time series for grass and other woody green species (woody sp.) found in the savanna understory at the Howard Springs OzFlux site from December 2012 to October 2014. The GCC daily data are shown with an 8-day centred running mean (marked every 8 days for visualisation) applied. The standard error of the mean is given by the shading. The GCC time series represent the change in relative greenness of grass and woody species, not the absolute sum of grass versus woody species biomass in the understory.





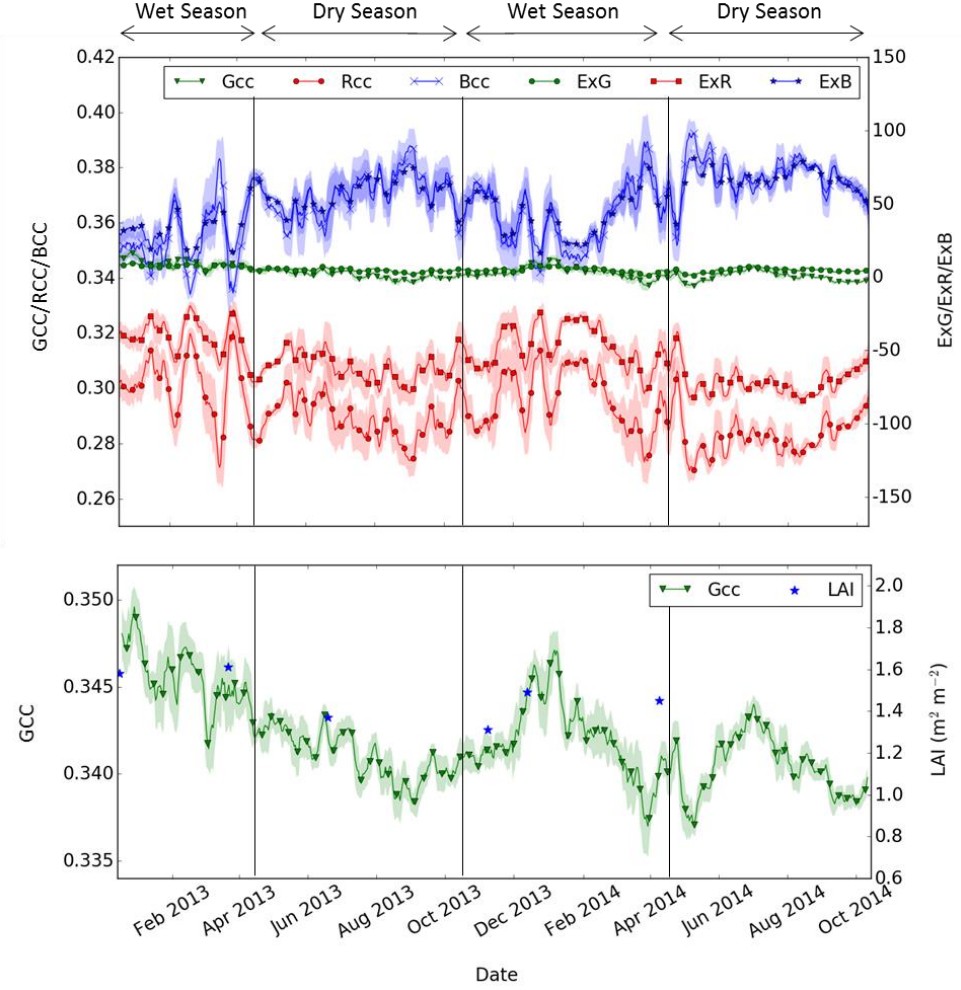

**Figure 6:** Daily green, red and blue chromatic coordinates (GCC/RCC/BCC) and excess indices (ExG/ExR/ExB) for the Howard Springs OzFlux site overstory (a), plus GCC and leaf area index (LAI) for the overstory (b) from December 2012 to October 2014. Daily data are shown with an 8-day centred running mean (marked every 8 days for visualisation) applied. The standard error of the mean is given by the shading.





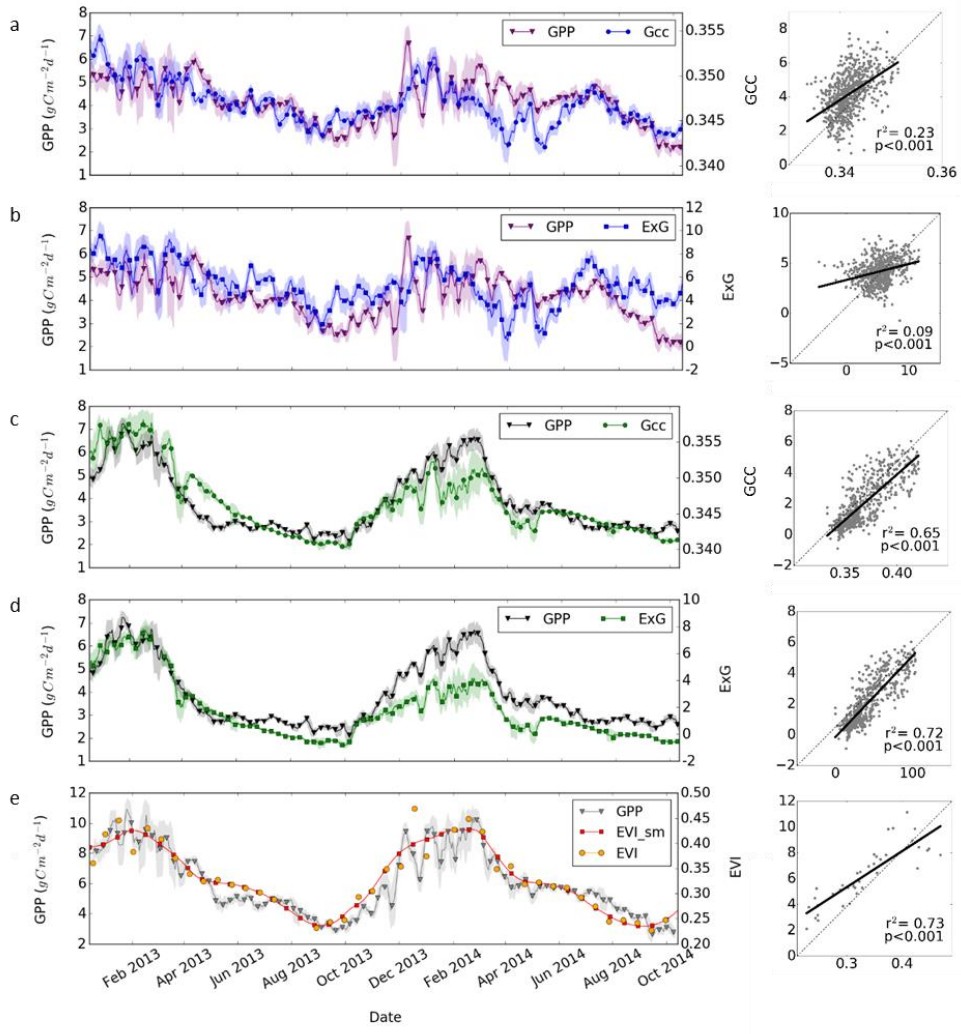

**Figure 7:** Overstory (a & b) and understory (c & d) flux tower GPP with green chromatic coordinate (GCC) and excess green (ExG) indices, as well as ecosystem flux tower GPP with MODIS enhanced vegetation index (EVI, e), from December 2012 to October 2014 at the Howard Springs OzFlux site. Daily data are shown with an 8-day running mean (marked every 8 days for visualisation) applied. The standard error of the mean is given by the shading. Included for each time series are the respective regression plots showing $r^2$ and p values for GCC/ExG/EVI (x) against flux tower GPP (y). For MODIS EVI (e) the time series plot includes raw 16 day values (EVI) and a Savitzky-Golay smoothed daily EVI product (EVI_sm), with the regression plot showing the raw 16 day EVI and the corresponding GPP for that day.



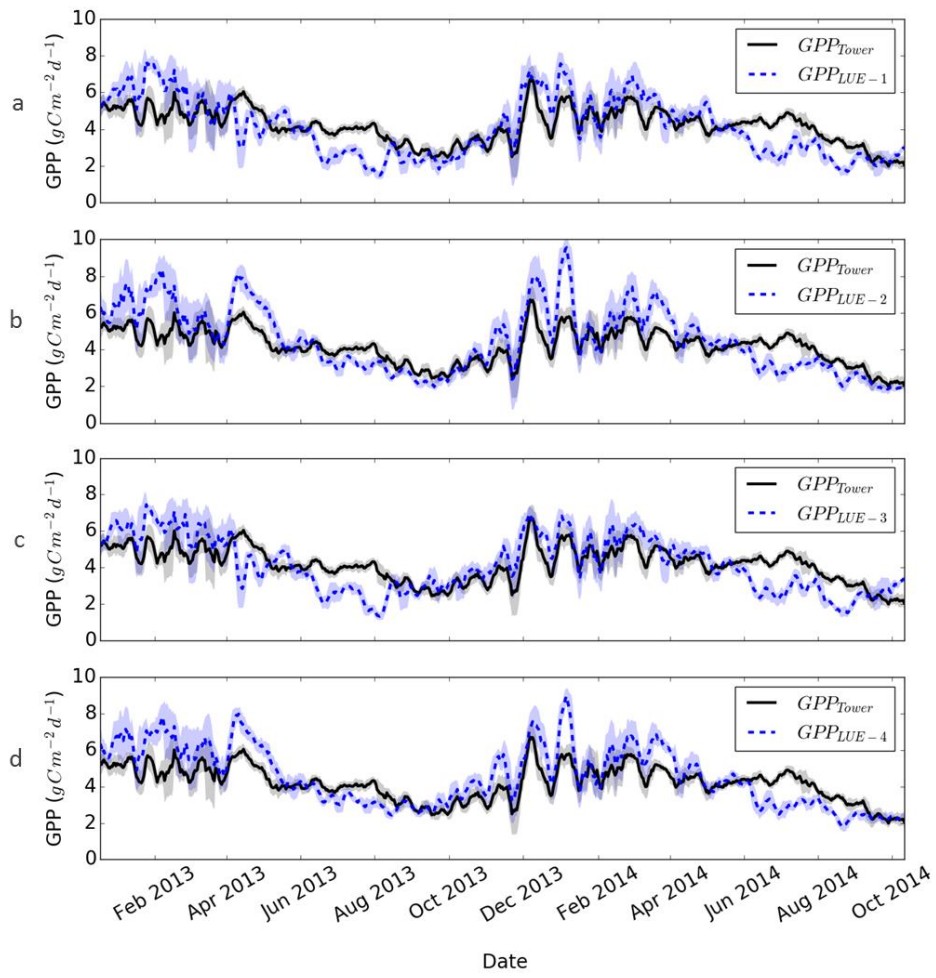

**Figure 8:** Overstory flux tower estimated GPP with model predicted GPP for the Howard Springs OzFlux site. Models shown are a) light use efficiency (LUE-1), b) LUE with evaporative fraction (LUE-2), c) LUE with green chromatic coordinates (LUE-3), d) and LUE with EF and GCC (LUE-4).





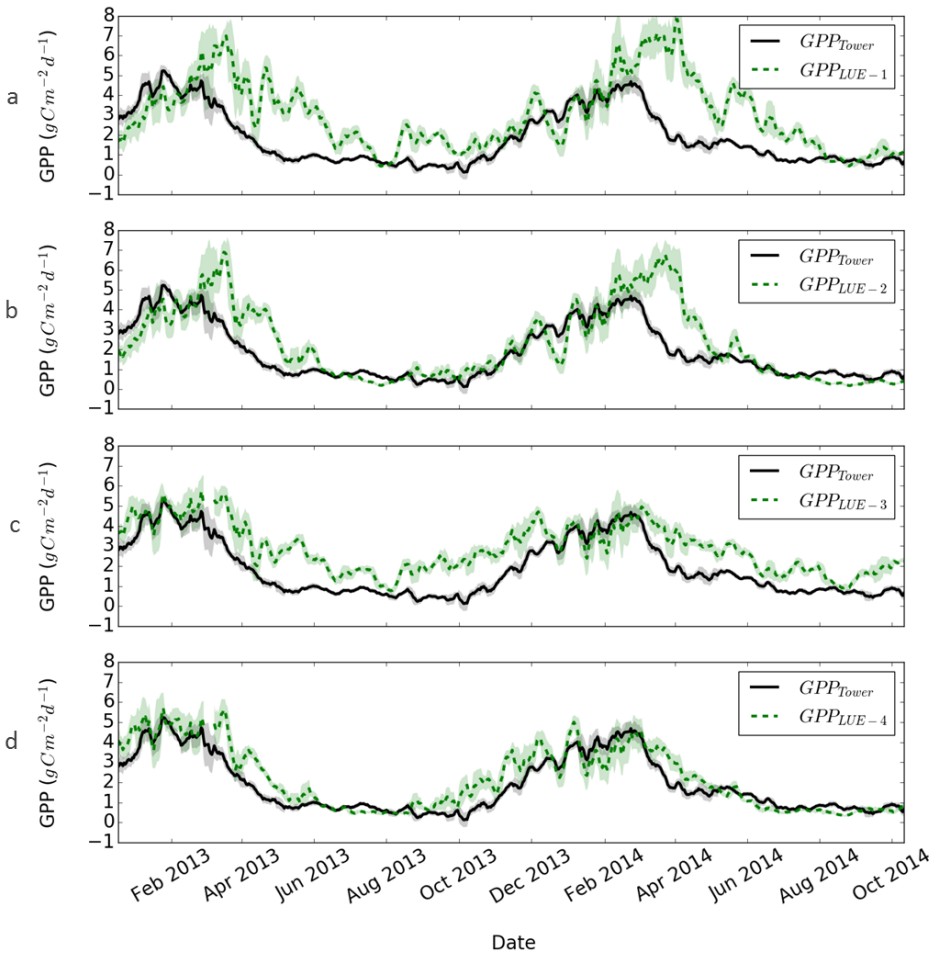

**Figure 9:** Understory flux tower estimated GPP with model predicted GPP for the Howard Springs OzFlux site. Models shown are a) light use efficiency (LUE-1), b) LUE with evaporative fraction (LUE-2), c) LUE with green chromatic coordinates (LUE-3), d) and LUE with EF and GCC (LUE-4).





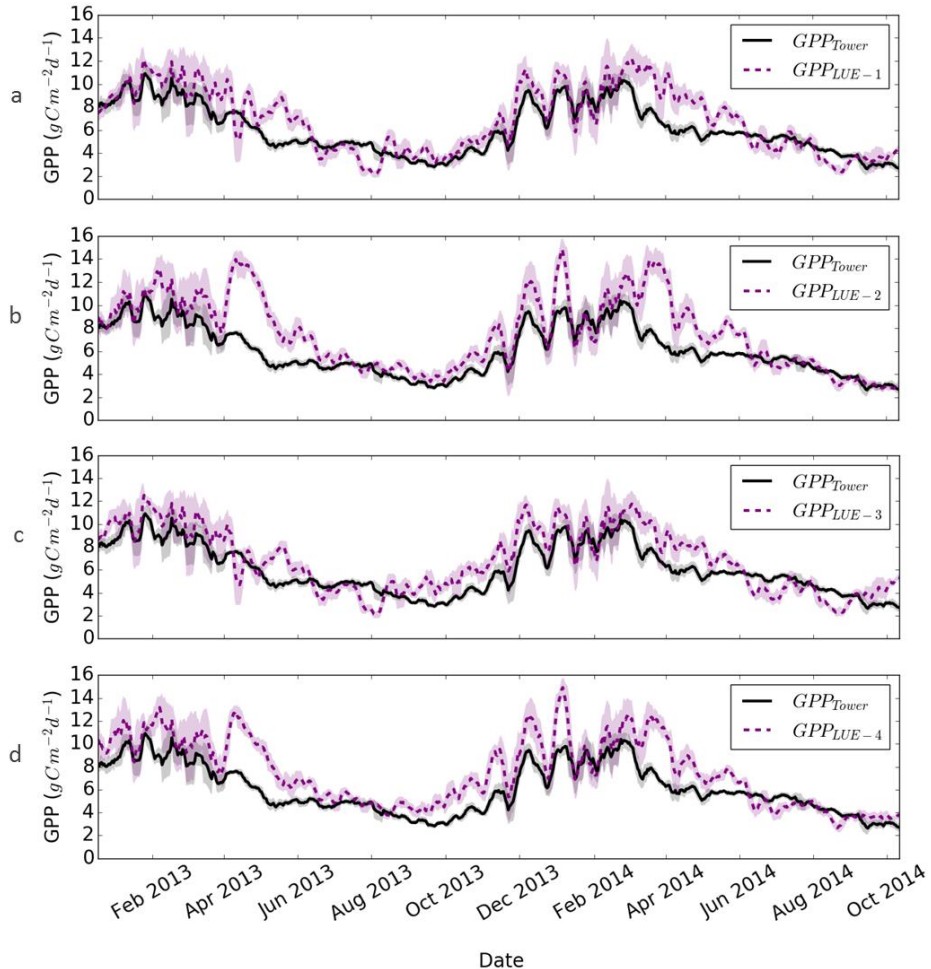

**Figure 10:** Ecosystem flux tower estimated GPP with model predicted GPP for the Howard Springs OzFlux site. Models shown are a) light use efficiency (LUE-1), b) LUE with evaporative fraction (EF, LUE-2), c) LUE with MODIS enhanced vegetation index (EVI, LUE-3), d) and LUE with EF and EVI (LUE-4).