# Peer review of "efficiency modelling of gross primary productivity for an"

_Biogeosciences, 2016_

## Referee Comment (RC1)

**Comments to the Authors:**

**Review of "Tree-grass phenoloy information improves the light use efficiency modelling of gross primary productivity for an Australian tropical Savanna".**

Firstly, I would like to congratulate the authors to a very interesting and well written manuscript. I truly enjoyed reading it and I learned a lot. However I have some questions that I would like to get answered before I can recommend this manuscript for publication:

This manuscript is not focusing on the EC based understory estimates of $CO_2$ fluxes, but I am still confused. The eddy covariance method is based on the assumption that measurements are done in the inertial surface layer, i.e. in the layer within the atmosphere where there are no vertical changes in fluxes depending on height of the sensors (Foken, 2008). This is not the case inside a canopy. Inside a canopy turbulence is very chaotic, and turbulent transport is much more efficient than above a canopy (Denmead and Bradley, 1987; Foken, 2008; Kaimal and Finnigan, 1994; Raupach, 1989). Additionally, there are sinks and sources in all directions in space. Fluxes can thereby basically come from any direction; the NEE estimated by the EC system is thereby not only a result of fluxes from the understory, it can equally as well be a result of respiration or CO2 uptake by the canopy cover above the EC system.

(P8 L20) In case you do not include the reflected PAR in the estimates of fraction of absorbed PAR (FAPAR), it is not FAPAR, it is the fraction of intercepted PAR (FIPAR). This is not the same thing. Generally, FIPAR is much more stable over the seasons than FAPAR, and this can make a difference in the estimate of the seasonal variation in GPP. Why did the reflected PAR data result in negative values during the dry season? It indicates some issues with the calibration of the sensors. Is there no way to inter-calibrate the sensors and recalculate the data? FAPAR is generally estimated as:

$$\text{FAPAR} = \frac{\text{PAR}_{in} - \text{PAR}_{ref} - (1-\alpha) \times \text{PAR}_{tr}}{\text{PAR}_{in}}$$

Where $\text{PAR}_{in}$ is incoming photosynthetic active radiation (PAR), $\text{PAR}_{ref}$ is reflected PAR, $\alpha$ is PAR albedo of the soil, and $\text{PAR}_{tr}$ is PAR transmitted through the vegetation.

I do not understand how the model can overestimate the GPP? You estimate a maximum LUE based on an average LUE for Dec-Mar. Then you use scalars with a value of between 0 and 1 to downscale the maximum LUE to a lower value. But since maximum LUE is based on the same time series of GPP as you use for the evaluation, it should not be possible for modelled GPP to be overestimated. Or did I misunderstand something? Please clarify.

**Specific comments:**

L11, it sounds like all grass in savannas is C4 species, which is absolutely not the case. Please just rephrase a bit.

P6 L1 Please describe very shortly the partitioning method used. Was it based on a light response curve or night time NEE-temperature curves?

Generally in the method section there are very many technical details. These are nice to have, but I think they could be moved to supplementary material to ease the reading of the manuscript. But, it is ok the way it is now as well, it is just a suggestion.

P9L18 APAR is in MJ **d-1**.

P5 please indicate the study period of the EC measurements, and other measurements by the way.

P9 L24 Why is n=8? In the figures it looks like the measurements started in January 2013, which would men n=7?

P9 L22 Why did you bin the LUE to months, this does not necessarily give the best indicator of maximum LUE. I would say that better would be to use a running mean for the estimates of seasonal dynamics in LUE, and then use the maximum value. Why should the average of 3 months give the best estimate for a maximum?

P16 L 17 Why did you use GCC as a proxy for FAPAR, and not as a scalar for LUE? There is strong seasonal variability in LUE depending on phenology of the vegetation, so I would think that it is more realistic to use the phenology as a direct scalar on LUE.

P10 L29 I assume that the regression was not used to replace APAR, but to replace FAPAR?

P12 L34 What limitations?

P13 L4 I would not consider a R2 value of 0.09 and 0.23 a well correlated relationship. These relationshis are not well correlated just because the p-value is significant. The assumptions for testing of significance is not fulfilled; there is high auto-correlation present in eddy covariance time series, so the true N is nowhere near the observed N. For example, Desai (2014) addresses this issue using a reduced degree of freedom calculation to show that the vast majority of flux tower regression is actually over-confident.

Fig 8-10. I suggest to incorporate subplots just like you did in Fig 7. Where you include a subplot with modelled GPP on the y-axis and the measured GPP on the x-axis. This really helps to see how well the models perform.

P15 L25-L27 Are you certain that RMSE is higher for the GCC included model (RMSE =1.43) than for the GCC and EF combined model(RMSE=1.36)? When looking at Fig 8 it does not look like RMSE can be higher. In Figure 8, it looks like the errors are much smaller; this should also be seen in the RMSE values.

**References:**

Desai, A.R., 2014. Influence and predictive capacity of climate anomalies on daily to decadal extremes in canopy photosynthesis. Photosynthesis Research, 119, 31-47, doi:10.1007/s11120-013-9925-z.

Denmead, O.T. and Bradley, E.F., 1987. On Scalar Transport in Plant Canopies. Irrigation Science, 8(2): 131-149.

Foken, T., 2008. Micrometeorology. Springer-Verlag, Berlin, 306 pp.

Kaimal, J.C. and Finnigan, J., 1994. Atmospheric boundary layer flows: their structure and measurement. Oxford University Press, Oxford.

Raupach, M.R., 1989. Applying Lagrangian fluid mechanics to infer scalar source distributions from concentration profiles in plant canopies. Agric. For. Meteorol., 47(2-4): 85-108.

---

## Referee Comment (RC2) · Anonymous Referee #2 · 13 Jun 2016

The paper "Tree-grass phenoloy information improves the light use efficiency modelling of gross primary productivity for an Australian tropical Savanna" is of scientific interest. Some issues (listed below) should be solved.

General comments

The use of GCC in the LUE model is thought to improve the GPP estimation because of the strong phenological cycle of the target. In my opinion the phenological cycle is very well represented when fAPAR is used. So the reason for using GCC must be different: replacing fAPAR measurements or testing if a "green" index (likely a proxy Printer-friendly version

of a "green" fAPAR) provides a better description of photosynthesis that that of total fAPAR.

Cameras pointing to trees: as large part of the ROI is occupied by the background (the sky), I wonder if the observed (and reduced) variability in GCC is not related to variations in sky optical properties during the year. The relation with LAI (Fig 6b) is not helping to figure it out, as the observed relation between GCC and LAI may be spurious (i.e., LAI increase and decrease in parallel to changes in sky optical properties). To disentangle the two effects it would be useful to define some additional ROIs with sky only and analyse the difference with the tree-ROIs selected. Performances of the different GPP models (4, all LUE based) are assessed in terms of r, RMSE and RPE. However, model 1 and 2 (eq 6 and use of EF) are used in prediction while (if I got it well) model 3 (using phenocam index) is in fitting (as two parameters, m and c coefficients) are adjusted. Model 4 (using MODIS) is in between, because a relationship is tuned between EVI and fAPAR. Therefore, results are not comparable in my opinion (see the discussion at page 15).

An interesting point is that the use of the phenocam index appears to eliminate the lag between measured and modelled GPP. The reason for this could be that the total fAPAR used by the other model is the source of this lag. On the contrary GCC may represent a kind of "green" fAPAR that is more in line with photosynthesis. A dedicated section comparing phenocam indexes and fAPAR would be very useful.

**Specific comments**

1 L 32 r2 ranging from 0.1 to 0.2 (overstory) is much lower than that of understory but they are both indicated as "well correlated".

3 L 23 I don't understand what is meant by "Core issues surrounding the remoteness of satellite sensors"

3 L23-25 this sentence is rather obscure ("the diffuse nature of light"?). I would sug-
gest to omit it and only mention that the highest temporal frequency available is one composite every 8-16 days.

3 L 34 I don't understand "via leaf emergence and senescence". Please rephrase.

4 L1-3 Here you are saying that LUE models describes GPP through the relation between APAR and LUE. There is no relation, they are both used to estimate GPP.

Section 2.3 The final field of view of the camera could be provided.

Section 2.3 Can you comment on possible effects of the automatic (and variable) white balance? This can variable from measure to measure. What is the effect on calculated indexes? Few numerical simulations may help in this assessment.

8 eq 16-18 Why is the reflected PAR is not used? This is fIPAR. And the resulting flux is IPAR not APAR 10 L 34 In which sense "predictive" is used here? Is there any validation / prediction on independent data (i.e. not used in fitting)?

Section 3.1 It would be interesting to see the FAPAR curves along with that of the various camera-indexes

Figure 7. Sorry, I am not getting what the 1:1 line refers to. The two variables on the scatterplot have different units and ranges

**Technical corrections**

3 L 4 Why "cover"?

7 L 5-7 This sentence says that it is homogeneous and it is not. It's a matter of scale. It can be rephrased.

8 L 13 "Absorbed" instead of "used".

11 L9 RCC/ExR looks like a ratio. I would suggest to use "and".

13 L1 I miss the integration in this section. The title of this section could be "Relation between GPP and time series of phenocam and MODIS indexes"

BGD
**P14 L5-7 Probably not needed, already described.**

---

## Referee Comment (RC3) · Anonymous Referee #3 · 27 Jul 2016

The utility of phenology information for improving GPP modeling results is an important research objective and I find the present work interesting and relevant. The paper is well written, methods are sound and results are carefully discussed. However, descriptions are generally very (too) detailed and several sections would benefit from a slightly more concise format. The structure of parts of the methods section should also be improved for improved overview, flow and clarity.

Some detailed and relatively minor comments:

1. Page 1 L32: An R2 of 0.09 – 0.23 does not constitute a well correlated relationship

as I see it.

2. Page 2 L16: I believe fire should be capitalized as in "..2015). Fire..."

3. Page 3 L19: What does the A2/A3 refer to? Is this information needed here?

4. Page 3 L20: MOD17 is mentioned to provide the most reliable means of estimating large-scale productivity. In comparison to what other products/estimates? MOD17 is known to be associated with significant uncertainty (related predominantly to the specification of the effective LUE), and I'm not convinced it will outperform other products given a full suite intercomparison.

5. Page 3 L23: "Core issues surrounding..."; Odd sentence. Suggest rewording. The full sentence structure (L23 to L28) should be rewritten for better language and clarity.

6. Section 2 introduction (Page 4): This intro piece doesn't outline the overall methodology well and/or the sub-division of the methods sections. I would probably leave it out completely or provide a more elaborate and cohesive piece.

7. Page 6 L2: I don't think that it is necessary to know the type of coding language (Python) used..

8. Page 6 L31: "f/stop"?

9. Sections 2.3 and 2.4: The methods are described in great detail. I would suggest reducing the wordiness as much as possible only including the most essential elements.

10. Section 2.4: I would include separate sub-sections for the phenocam and radiation data processing for improved flow and readability. Line 13 on page 8 could be the start of the LUE sub-section.

11. Page 7 L24-26: I feel that this information is redundant.

12. Page 8 L22: Shouldn't leaf absorptance be considered in the APAR calculation? You are using fPAR and not fAPAR, right?

13. Page 8 L24-: The information on LAI collection, clumping etc is out of place. You will need a separate section on this.

14. Page 10 L1-4: Is it valid to adopt the default MOD17 savanna values for your study site? Did you verify these against the tower observations?

15. Section 3.1 is very detailed and would benefit from a more concise format, if possible.

---

## Author Comment (AC1) · 24 Aug 2016

Authors' response to reviewer comments for manuscript bg-2016-187 "Moore et al., Tree-grass phenology information improves light use efficiency modelling of gross primary productivity for an Australian tropical savanna"

We wish to thank all three reviewers for their helpful and constructive comments regarding our manuscript. Their comments are relevant and we feel will improve our manuscript. Below we outline our response and the way in which we have addressed each of their comments.

Reviewer #1

Firstly, I would like to congratulate the authors to a very interesting and well written manuscript. I truly enjoyed reading it and I learned a lot. However I have some questions that I would like to get answered before I can recommend this manuscript for publication:

This manuscript is not focusing on the EC based understory estimates of CO2 fluxes, but I am still confused. The eddy covariance method is based on the assumption that measurements are done in the inertial surface layer, i.e. in the layer within the atmosphere where there are no vertical changes in fluxes depending on height of the sensors (Foken, 2008). This is not the case inside a canopy. Inside a canopy turbulence is very chaotic, and turbulent transport is much more efficient than above a canopy (Denmead and Bradley, 1987; Foken, 2008; Kaimal and Finnigan, 1994; Raupach, 1989). Additionally, there are sinks and sources in all directions in space. Fluxes can thereby basically come from any direction; the NEE estimated by the EC system is thereby not only a result of fluxes from the understory, it can equally as well be a result of respiration or CO2 uptake by the canopy cover above the EC system.

The understory EC data used in this study is that which is already published in the same special issue of Biogeosciences (Moore et al., 2016). This paper discusses and validates the use of an understory flux tower for savanna research in more detail. In particular, it presents results from a cospectral analysis, based on the work of Kaimal and Finnigan (1994) (as referenced by the reviewer), to show that the flux tower does primarily record vertical transport during daytime turbulent conditions. We referred to Moore et al. (2016) within section 2.2 to direct readers to this more detailed discussion. Given all three reviewers commented on the length of the manuscript, we feel it would be ineffective to elaborate on this further. However, we will ensure that more explicit reference is made to Moore et al. (2016) within section 2.2 in order to help resolve some of this confusion.

(P8 L20) In case you do not include the reflected PAR in the estimates of fraction of absorbed PAR (FAPAR), it is not FAPAR, it is the fraction of intercepted PAR (FIPAR). This is not the same thing. Generally, FIPAR is much more stable over the seasons than FAPAR, and this can make a difference in the estimate of the seasonal variation in GPP. Why did the reflected PAR data result in negative values during the dry season? It indicates some issues with the calibration of the sensors. Is there no way to intercalibrate the sensors and recalculate the data? FAPAR is generally estimated as:

FAPAR = PARin – PARref –  $(1 - \alpha)$  x PARtr PARin

Where PARin is incoming photosynthetic active radiation (PAR), PARref is reflected PAR,  $\alpha$  is PAR albedo of the soil, and PARtr is PAR transmitted through the vegetation.

The reviewer is correct here and given reviewer 2 and 3 also highlighted this point, it prompted us to re-check our analysis of fPAR and APAR.

Initially the reason for omitting reflected PAR was due to fPAR values often being negative in the understory in the late dry season. This was most likely due to the lack of vegetation in the understory in the late dry season around some of the towers, which caused incoming PAR below the understory to be almost equal to that of PAR above the understory. For one tower, PAR below the understory was higher than PAR above the understory, which is a result of the heterogeneous nature of the savanna ecosystem at these point scales. By omitting this tower from the analysis during the late dry season, negative fPAR values no longer occurred in the understory. This data was then used to calculate APAR, not IPAR.

Basically, there was a bug in the code that was missed on previous checks before submission. This bug was due to an incorrect labelling of the APAR variable to an alternative version, which omitted the reflected/upwelling PAR to test the above theory about the negative fPAR values. Therefore, we incorrectly concluded that by omitting the reflected PAR, the model performed better, when in fact it was actually using the correct, reflected PAR-included APAR values.

СЗ

To sum up, we are grateful for the keen eyes of all three reviewers here for picking up on this mistake before the manuscript made it further in the review process. Thankfully, the data presented are correct, they were just interpreted incorrectly on our behalf, so we will amend this in the resubmission.

I do not understand how the model can overestimate the GPP? You estimate a maximum LUE based on an average LUE for Dec-Mar. Then you use scalars with a value of between 0 and 1 to downscale the maximum LUE to a lower value. But since maximum LUE is based on the same time series of GPP as you use for the evaluation, it should not be possible for modelled GPP to be overestimated. Or did I misunderstand something? Please clarify.

We are a little unsure as to what the reviewer is referring to with this statement, if it is one aspect of the text/figure or if it is our general approach to our research question. However, to answer this query at a general level, the model can most definitely overestimate GPP (or underestimate it) as LUE is not the only input to the GPP model. APAR is also an input, which in the case of the savannas is often overestimated during the transition periods between wet and dry seasons (i.e., Kanniah et al., 2009, Whitley et al., 2011). Meteorology also drives the down-regulation of maximum LUE to daily LUE variability, so although we obtained maximum (peak) LUE from our GPP estimates, the application of this down-regulation process means the two parameters are no longer directly related. Therefore, by using APAR and LUE in the model, GPP can be over- or underestimated. This is why we chose to test whether including phenology information would improve the model's ability to capture flux tower GPP, given this savanna ecosystem displays such a distinct boom-bust seasonal phenology.

Specific comments: L11, it sounds like all grass in savannas is C4 species, which is absolutely not the case. Please just rephrase a bit.

We will fix this in the line identified

P6 L1 Please describe very shortly the partitioning method used. Was it based on a light response curve or night time NEE-temperature curves?

We used a u\* filter and artificial neural network approach, with soil water, soil temperature, air temperature and EVI as the main model drivers, to determine respiration (R), assuming all night time NEE was R. This was extrapolated to the daytime and GPP was calculated as the difference between R and NEE. Further information about this process can be found in Beringer et al. (2016), also an article in the special issue our manuscript is a part of. Hence, we will add this short description to P6, L1 and direct the reader to Beringer et al. (2016) for further information.

Generally in the method section there are very many technical details. These are nice to have, but I think they could be moved to supplementary material to ease the reading of the manuscript. But, it is ok the way it is now as well, it is just a suggestion.

This is a good suggestion, and given the other reviewers have also suggested this, we will revise the manuscript and shorten where possible.

P9L18 APAR is in MJ d-1.

We will fix this in the text.

P5 please indicate the study period of the EC measurements, and other measurements by the way.

The study period was from 12th December 2012 to 14th October 2014, for all measurements. We will add this to the text.

P9 L24 Why is n=8? In the figures it looks like the measurements started in January 2013, which would men n=7?

N=8 because it includes the months of Dec through to Mar (inclusive), which each occur twice during the study period. We stated this in the text on P9, L22, but could make it clearer in the revised version.

P9 L22 Why did you bin the LUE to months, this does not necessarily give the best indicator of maximum LUE. I would say that better would be to use a running mean for the estimates of seasonal dynamics in LUE, and then use the maximum value. Why should the average of 3 months give the best estimate for a maximum?

We binned LUE by month and termed it peak LUE, rather than maximum LUE, because true maximum LUE is not easy to obtain from EC measurements. What we wanted to get at was a representative maximum LUE that was obtained during conditions that were not limiting to growth. A similar approach was used by Kanniah et al. (2009), so we intended to mirror their approach in terms of calculating a maximal LUE estimate from EC measurements. Perhaps better would be to describe LUE as "optimum" rather than "peak", as peak suggests it is an instantaneous maximum rather than an overall optimum value.

P16 L 17 Why did you use GCC as a proxy for FAPAR, and not as a scalar for LUE? There is strong seasonal variability in LUE depending on phenology of the vegetation, so I would think that it is more realistic to use the phenology as a direct scalar on LUE.

We used GCC as a proxy for fPAR because the high values of fPAR in the transition periods were what we believed to be the source of the error in the model. LUE reduces rapidly from Feb to May, which is more characteristic of the phenology response seen in the field (i.e. Figure 3). Given this, the LUE was more indicative of phenology driven GPP than APAR, so was less likely to be the source of the error in the model than APAR.

P10 L29 I assume that the regression was not used to replace APAR, but to replace

**FAPAR?**

The phrasing of this sentence is misleading. It should read "Daily EVI were regressed against site-based daily ecosystem fPAR, and the regression was used along with incoming PAR information to replace APAR in Eq. (6)."

**P12 L34 What limitations?**

The limitations refer to those mentioned in line 29 of the same paragraph. Perhaps better would be to refer the above mentioned uncertainty. The suggested additional analysis from reviewer 2, once added here, should also help resolve this confusion.

P13 L4 I would not consider a R2 value of 0.09 and 0.23 a well correlated relationship. These relationshis are not well correlated just because the p-value is significant. The assumptions for testing of significance is not fulfilled; there is high auto-correlation present in eddy covariance time series, so the true N is nowhere near the observed N. For example, Desai (2014) addresses this issue using a reduced degree of freedom calculation to show that the vast majority of flux tower regression is actually overconfident.

We agree with the reviewer here, better would be to say more broadly that the relationship was stronger for the understory than for the overstory. We will change this in the text accordingly. \_\_\_\_\_

Fig 8-10. I suggest to incorporate subplots just like you did in Fig 7. Where you include a subplot with modelled GPP on the y-axis and the measured GPP on the x-axis. This really helps to see how well the models perform.

This is a good idea. The reason we did not do it from the beginning was because we felt it made the figures too busy, so we included this information in Table 2 instead. However, we do agree that it would add to the figures, so we can include it in our

**resubmission.**

We can see what the reviewer means in regards to the RMSE values in Table 2 vs. the timeseries in Figure 8. We have double checked our values and have found that some values in Table 2 need amending, which we will do in the resubmission. However, for the overstory, the RMSE value is lower for the LUE\_EF\_GCC model when compared with the LUE\_GCC model, despite what Figure 8 indicates. What we think has caused this is that while the error appears enhanced in the wet season in the LUE\_EF\_GCC combined model, it is reduced in the dry season, which results in RMSE being slightly lower than for the GCC model alone. By adding the scatterplots to each of these figures, as suggested by reviewer 1, this may help to reduce some of this confusion.

**Reviewer #2**

P15 L25-L27 Are you certain that RMSE is higher for the GCC included model (RMSE =1.43) than for the GCC and EF combined model(RMSE=1.36)? When looking at Fig 8 it does not look like RMSE can be higher. In Figure 8, it looks like the errors are much smaller; this should also be seen in the RMSE values.

The use of GCC in the LUE model is thought to improve the GPP estimation because of the strong phenological cycle of the target. In my opinion the phenological cycle is very well represented when fAPAR is used. So the reason for using GCC must be different: replacing fAPAR measurements or testing if a "green" index (likely a proxy of a "green" fAPAR) provides a better description of photosynthesis that that of total fAPAR.

The reviewer raises a valid point here, in that fAPAR does capture the phenological cycle reasonably well. However, it does not capture it perfectly and is particularly poor during the transition from the wet to dry season (or dry to wet). We believe this is due to the senescence of the understory grasses that changes the greenness and

GPP of the savanna despite fPAR remaining high. Currently, savanna productivity models poorly capture this change (i.e., Kanniah et al., 2009, Whitley et al., 2011, Whitley et al., 2016), and we would argue it is because they do not capture the understory phenology dynamics as well as they could. Moore et al. (2016) found that the understory accounts for 1/3 of savanna GPP, which is heavily dominated by the annual grasses that show this strong phenology. When models only use the fPAR(or APAR) information, they fail to capture the transition from wet to dry (and dry to wet) and over-estimate GPP. By using the GCC information, which provides a more accurate representation of phenology when compared with APAR for this savanna, the LUE model performs better. Ma et al. (2014) also reached a similar conclusion when they used EVI to incorporate better phenology information into their GPP model.

Cameras pointing to trees: as large part of the ROI is occupied by the background (the sky), I wonder if the observed (and reduced) variability in GCC is not related to variations in sky optical properties during the year. The relation with LAI (Fig 6b) is not helping to figure it out, as the observed relation between GCC and LAI may be spurious (i.e., LAI increase and decrease in parallel to changes in sky optical properties). To disentangle the two effects it would be useful to define some additional ROIs with sky only and analyse the difference with the tree-ROIs selected.

This is a really good idea and we will endeavour to analyse a sky-only ROI to compare against our overstory estimates. We will include a discussion of these results in the revised manuscript and add the sky timeseries to support figure 6.

Performances of the different GPP models (4, all LUE based) are assessed in terms of r, RMSE and RPE. However, model 1 and 2 (eq 6 and use of EF) are used in prediction while (if I got it well) model 3 (using phenocam index) is in fitting (as two parameters, m and c coefficients) are adjusted. Model 4 (using MODIS) is in between, because a relationship is tuned between EVI and fAPAR. Therefore, results are not comparable in

my opinion (see the discussion at page 15).

Each model combination is compared against flux tower-derived GPP estimates and the r, RMSE and RPE provide an indication of which model is best at capturing tower GPP. Our discussion on pg 15 discusses which of the model combinations was best at capturing tower GPP, finding that the inclusion of phenology information did the best job.

An interesting point is that the use of the phenocam index appears to eliminate the lag between measured and modelled GPP. The reason for this could be that the total fAPAR used by the other model is the source of this lag. On the contrary GCC may represent a kind of "green" fAPAR that is more in line with photosynthesis. A dedicated section comparing phenocam indexes and fAPAR would be very useful.

We can address these concepts in the discussion, which we feel will also help to address the queries raised by this reviewer about the use of GCC as a better indicator of phenology in the model, as identified above.

Specific comments:

1 L 32 r2 ranging from 0.1 to 0.2 (overstory) is much lower than that of understory but they are both indicated as "well correlated".

Agreed, we will amend our statement in the text, as per our response to reviewer 1 above.

Here we meant to identify that one of the limitations of satellite remote sensors is their remoteness from the ecosystems they measure. We will re-phrase the sentence to

3 L 23 I don't understand what is meant by "Core issues surrounding the remoteness of satellite sensors"

state this more clearly.

3 L23-25 this sentence is rather obscure ("the diffuse nature of light"?). I would suggest to omit it and only mention that the highest temporal frequency available is one composite every 8-16 days.

We will omit the section identified by the reviewer so it reads more concisely.

3 L 34 I don't understand "via leaf emergence and senescence". Please rephrase.

This sentence is talking about the value of phenocams for identifying leaflevel changes, such as leaf emergence and senescence, so we will rephrase the sentence to more clearly show this. In particular, we will remove the word "via" as this seems to be the most misleading part of the sentence.

4 L1-3 Here you are saying that LUE models describes GPP through the relation between APAR and LUE. There is no relation, they are both used to estimate GPP.

Here we used the word 'relation' to indicate that the two parameters were multiplied to obtain GPP. We will reword this sentence to be clearer about this.

Section 2.3 The final field of view of the camera could be provided.

We will provide this in the section identified.

Section 2.3 Can you comment on possible effects of the automatic (and variable) white balance? This can variable from measure to measure. What is the effect on calculated indexes? Few numerical simulations may help in this assessment.

We do discuss, albeit briefly, the effects of white balance on image collection in the limitations section of our manuscript. The reviewer is correct in their assessment that white balance can vary from image to image, which is particularly more prevalent during lower sun angles i.e. dawn/dusk. By using middle of day values, the effects of white balance can be reduced. However, white balance was set to zero in our analysis, which is a limitation in that it increases the scene illumination noise in our images. However, given that we only analysed middle of day images in an environment that is highly dynamic, the phenology signal was still identifiable. This may not be the case for a less dynamic ecosystem. Migliavacca et al. (2011) discuss the uncertainty and limitations of using digital camera imagery, which we make reference to in section 3.4. However, we will provide some additional discussion around this point in section 3.4 in order to address this comment by the reviewer.

8 eq 16-18 Why is the reflected PAR is not used? This is fIPAR. And the resulting flux is IPAR not APAR

Our answer to reviewer 1 about this should help to clarify this point.

10 L 34 In which sense "predictive" is used here? Is there any validation / prediction on independent data (i.e. not used in fitting)?

The relative 'predictive' error indicator we used in our analysis is simply a calculation of the % mean difference between two datasets. It provides an indication of the direction of change in the predicted values relative to the measured values in a relative sense. See Kanniah et al. (2009) Appendix 1 for further explanation and formulas for calculation.

Section 3.1 It would be interesting to see the FAPAR curves along with that of the various camera-indexes

This is a nice idea, but in the interest of balancing the additional information requests and the length of the manuscript in its current form, we think that creating and discussing an additional plot would make the manuscript unwieldly. However, we are happy to add these at the editors discretion and suggest that if the editor advises them, we include them as supplementary information.

Figure 7. Sorry, I am not getting what the 1:1 line refers to. The two variables on the scatterplot have different units and ranges

We can see why this would be confusing, it was meant to simply provide a guide of the deviation of the data, so we will remove the line from the resubmitted version.

Technical corrections:

3 L 4 Why "cover"?

We can remove the word 'cover' in the sentence to simply read "phenological change"

7 L 5-7 This sentence says that it is homogeneous and it is not. It's a matter of scale. It can be rephrased.

This is absolutely true, but the sentence in question does state this: "While the understory is largely homogenous in species distribution at the flux tower scale (i.e. >50 m), variation from one point to another does exist in the understory due to its vegetation composition." The sentence could be amended to read "...variation does exists at the smaller scale (i.e. < 5 m) in the understory due to..." to be a bit clearer on the subject.

8 L 13 "Absorbed" instead of "used".

We will fix this in the revised version.

11 L9 RCC/ExR looks like a ratio. I would suggest to use "and".

We will fix this in the revised version. \_

13 L1 I miss the integration in this section. The title of this section could be "Relation between GPP and time series of phenocam and MODIS indexes"

This section is about using the phenology information to improve estimates of GPP. Given this, we agree with the reviewer that the heading is misleading, therefore we propose to change it to "Using phenocam and MODIS phenology to improve GPP model estimates".

P14 L5-7 Probably not needed, already described.

We will remove this sentence from the text.

**Reviewer #3**

The utility of phenology information for improving GPP modeling results is an important research objective and I find the present work interesting and relevant. The paper is well written, methods are sound and results are carefully discussed. However, descriptions are generally very (too) detailed and several sections would benefit from a slightly more concise format. The structure of parts of the methods section should also be improved for improved overview, flow and clarity.

We are pleased the reviewer enjoyed our manuscript and do agree that it is quite lengthy in parts. Given we used a rather home-made camera for our phenocams, we felt we should provide more detail about our methods. However, we will revise our manuscript and remove less-relevant information where possible. Alternatively, we can formulate a supplementary materials file that documents the camera setup in finer detail, and provide only a brief overview in the methods section of the manuscript.

Some detailed and relatively minor comments:

1. Page 1 L32: An R2 of 0.09 - 0.23 does not constitute a well correlated relationship

as I see it.

This was also identified by reviewers 1 & 2, so we will fix this in the text as per our response to reviewer 1. \_\_\_\_\_

2. Page 2 L16: I believe fire should be capitalized as in "..2015). Fire: : :"Yes it should, we will fix this in the revised manuscript.

3. Page 3 L19: What does the A2/A3 refer to? Is this information needed here?

The A2/A3 information refers to the sub-product of MOD17 used, as it is a combination of both GPP (A2) and NPP (A3) obtained from the Terra satellite. Given we only used the MOD17 A2 (i.e. GPP) product, we can omit the A3 reference, but feel the A2 reference should be kept for clarity.

4. Page 3 L20: MOD17 is mentioned to provide the most reliable means of estimating large-scale productivity. In comparison to what other products/estimates? MOD17 is known to be associated with significant uncertainty (related predominantly to the specification of the effective LUE), and I'm not convinced it will outperform other products given a full suite intercomparison.

We agree with the reviewer here in that there are a suite of GPP model products available. However, it is out of the scope of our study to compare all products. This sentence should therefore be amended to remove the "most reliable" portion with something reading "...the MODIS GPP product is widely-used means of estimating..." instead.

5. Page 3 L23: "Core issues surrounding: : :"; Odd sentence. Suggest rewording. The full sentence structure (L23 to L28) should be rewritten for better language and clarity.

This statement was also identified by reviewer 2, so we will fix the sentence based on our response provided previously.

6. Section 2 introduction (Page 4): This intro piece doesn't outline the overall methodology well and/or the sub-division of the methods sections. I would probably leave it out completely or provide a more elaborate and cohesive piece.

The intention of this short section was to provide a brief overview/blurb of the methods before describing what was done. Given the reviewers all commented on the length of our manuscript, we will omit it in the resubmission.

7. Page 6 L2: I don't think that it is necessary to know the type of coding language (Python) used..

We will remove this from the section identified.

8. Page 6 L31: "f/stop"?

This is a photography term that refers to the ratio of a lens' focal length to the diameter of the point where light enters the camera. It can be referred to as a focal point. We didn't feel it was necessary to describe it but could add "(focal point)" after it in the text.

Eddy covariance in itself should really have a more detailed explanation than the one

9. Sections 2.3 and 2.4: The methods are described in great detail. I would suggest reducing the wordiness as much as possible only including the most essential elements.

We will endeavour to reduce the methods section as much as possible without losing the necessary information required to repeat the science. It is lengthy, but the paper also includes a range of measurement that need to be properly described.

While this reviewer has suggested we shorten the methods, the other two reviewers have encouraged us to provide more detail on some aspects of analysis (i.e. EC measurements, phenocam processing).

given in our manuscript. Likewise, given the phenocams we used were a modified point-and-shoot digital camera, we needed to provide enough information about how they were set up and used in order for scientific replication. Similarly for the PAR, LAI and LUE model data.

10. Section 2.4: I would include separate sub-sections for the phenocam and radiation data processing for improved flow and readability. Line 13 on page 8 could be the start of the LUE sub-section.

This is a great suggestion and we will split the section where indicated by the reviewer.

11. Page 7 L24-26: I feel that this information is redundant.

We will remove this information in the re-submitted manuscript.

12. Page 8 L22: Shouldn't leaf absorptance be considered in the APAR calculation? You are using fPAR and not fAPAR, right?

Our response to reviewer 1 regarding this should help clarify this point.

13. Page 8 L24-: The information on LAI collection, clumping etc is out of place. You will need a separate section on this.

We will also separate this section into a new subsection in the methods.

14. Page 10 L1-4: Is it valid to adopt the default MOD17 savanna values for your study site? Did you verify these against the tower observations?

The Tmin and VPD values were previously validated for the Howard Springs site by Kanniah et al. (2009). However, we found slightly higher maximum VPD for our study period than that of Kanniah et al. (2009). Therefore, we cited the original values of Running & Zhao (2015) for our study. Given Kanniah et al. (2009) did perform a validation of earlier values of Running et al. (2006) for savannas, we will include Kanniah et al. (2009) in our citation of the section identified.

15. Section 3.1 is very detailed and would benefit from a more concise format, if possible.

We will endeavour to shorten this section where possible in the resubmission.

References Cited:

BERINGER, J., MCHUGH, I., HUTLEY, L. B., ISAAC, P. & KLJUN, N. 2016. Dynamic INtegrated Gap-filling and partitioning for OzFlux (DINGO). Biogeosciences Discuss., 2016, 1-36. KAIMAL, J. C. & FINNIGAN, J. J. 1994. Atmospheric boundary layer flows: their structure and measurement, New York, Oxford University Press. KANNIAH, K. D., BERINGER, J., HUTLEY, L. B., TAPPER, N. J. & ZHU, X. 2009. Evaluation of Collections 4 and 5 of the MODIS Gross Primary Productivity product and algorithm improvement at a tropical savanna site in northern Australia. Remote Sensing of Environment, 113, 1808-1822. MA, X., HUETE, A., YU, Q., RESTREPO-COUPE, N., BERINGER, J., HUTLEY, L. B., KANNIAH, K. D., CLEVERLY, J. & EAMUS, D. 2014. Parameterization of an ecosystem light-use-efficiency model for predicting savanna GPP using MODIS EVI. Remote Sensing of Environment, 154, 253-271. MIGLIAVACCA, M., GALVAGNO, M., CREMONESE, E., ROSSINI, M., MERONI, M., SONNENTAG, O., COGLIATI, S., MANCA, G., DIOTRI, F., BUSETTO, L., CESCATTI, A., COLOMBO, R., FAVA, F., MORRA DI CELLA, U., PARI, E., SINISCALCO, C. & RICHARDSON, A. D. 2011. Using digital repeat photography and eddy covariance data to model grassland phenology and photosynthetic CO2 uptake. Agricultural and Forest Meteorology, 151, 1325-1337. MOORE, C. E., BERINGER, J., EVANS, B., HUTLEY, L. B., MCHUGH, I. & TAPPER, N. J. 2016. The contribution of trees and grasses to productivity of an Australian tropical savanna. Biogeosciences, 13, 2387-2403. WHITLEY, R., BERINGER, J., HUTLEY, L. B., ABRAMOWITZ, G., DE KAUWE, M. G., EVANS, B., HAVERD, V., LI, L., MOORE, C., RYU, Y., SCHEITER, S., SCHYMANSKI, S. J., SMITH, B., WANG, Y. P., WILLIAMS, M. & YU, Q. 2016. Challenges and opportunities in modelling savanna ecosystems. Biogeosciences Discuss., 2016, 1-44. WHITLEY, R. J., MACINNIS-NG, C. M. O., HUTLEY, L. B., BERINGER, J., ZEPPEL, M., WILLIAMS, M., TAYLOR, D. & EAMUS, D. 2011. Is productivity of mesic savannas light limited or water limited? Results of a simulation study. Global Change Biology, 17, 3130-3149.

---

## Author Response (AR1)

Authors' response to reviewer comments for manuscript bg-2016-187 *"Moore et al., Tree-grass phenology information improves light use efficiency modelling of gross primary productivity for an Australian tropical savanna"*

We wish to thank all three reviewers, and Dr. Migliavacca, for their helpful and constructive comments regarding our manuscript. Their comments are relevant and we feel have improved our manuscript. Below we outline our response and the way in which we have addressed each of their comments. Some additional pieces of analysis have been provided in a supplementary materials file in order to balance manuscript length with our response to their suggestions.

Dear authors,

The reviewers of the manuscript "Tree-grass phenology information improves light use efficiency modelling of gross primary productivity for an Australian tropical savanna" recognized the work as relevant for the audience of Biogeosciences, so do I. The Reviewers' suggested a series of additional analysis and editorial modifications to improve both the readability and the robustness of the analysis. In the Interactive Comment the authors nicely discuss how the reviewers' comments can be incorporated in the revised manuscript. Therefore, the manuscript should reconsidered after the major revisions suggested by the Reviewers' will be included.
One contrasting comments from the reviewers was about the trade off between length of the paper and methodology details provided. I agree with the authors consideration that, although the paper is lengthy, it needs to include a range of measurement that need to be properly described for sake of repeatability. The use of supplementary materials can be an option.
During the review I suggest the authors to address the issue raised by Reviewer 1 about the calculation of APAR, which can also help to address the comments of Reviewer 2 and 3 about the time lag between GCC, APAR, and GPP, as well as the differences between GCC and APAR.
Also, the authors to discuss more deeply the issue raised by Reviewer 1 about the estimation of the peak LUE. I consider valuable the idea of the Reviewer to use running means instead of the average by months. Another option for the authors is to optimize the peak LUE value.
The reviewers suggested also a valuable analysis to assess the robustness of the indices extracted from the digital camera data.
Last but not least the Reviewers pointed out a series of modification to Tables and Figures that need to be included.
Looking forward for reading the revised version.
Best Regards,
Mirco Migliavacca

**Reviewer #1**

*Firstly, I would like to congratulate the authors to a very interesting and well written manuscript. I truly enjoyed reading it and I learned a lot. However I have some questions that I would like to get answered before I can recommend this manuscript for publication:*

*This manuscript is not focusing on the EC based understory estimates of CO2 fluxes, but I am still confused. The eddy covariance method is based on the assumption that measurements are done in the inertial surface layer, i.e. in the layer within the atmosphere where there are no vertical changes in fluxes depending on height of the sensors (Foken, 2008). This is not the case inside a canopy. Inside a canopy turbulence is very chaotic, and turbulent transport is much more efficient than above a canopy (Denmead and Bradley, 1987; Foken, 2008; Kaimal and Finnigan, 1994; Raupach, 1989). Additionally, there are sinks and sources in all directions in space. Fluxes can thereby basically come from any direction; the NEE estimated by the EC system is thereby not only a result of fluxes from the understory, it can equally as well be a result of respiration or CO2 uptake by the canopy cover above the EC system.*

The understory EC data used in this study is that which is already published in the same special issue of Biogeosciences (Moore et al., 2016). This paper discusses and validates the use of an understory flux tower for savanna research in more detail. In particular, it presents results from a cospectral analysis, based on the work of Kaimal and Finnigan (1994) (as referenced by the reviewer), to show that the flux tower does primarily record vertical transport during daytime turbulent conditions. We referred to Moore et al. (2016) within section 2.2 to direct readers to this more detailed discussion. Given all three reviewers commented on the length of the manuscript, we feel it would be ineffective to elaborate on this further. However, we have expanded the sentence on page 5 that discusses the validation of the understory to now read: "*The understory tower primarily recorded vertical transfer during turbulent conditions, which was validated via power spectra analysis (Moore et al., 2016a) that followed idealised curves for vegetated canopies (Kaimal and Finnigan, 1994).*"
* * *
*(P8 L20) In case you do not include the reflected PAR in the estimates of fraction of absorbed PAR (FAPAR), it is not FAPAR, it is the fraction of intercepted PAR (FIPAR). This is not the same thing. Generally, FIPAR is much more stable over the seasons than FAPAR, and this can make a difference in the estimate of the seasonal variation in GPP. Why did the reflected PAR data result in negative values during the dry season? It indicates some issues with the calibration of the sensors. Is there no way to inter-calibrate the sensors and recalculate the data? FAPAR is generally estimated as:*

$$FAPAR = \frac{PAR_{in} - PAR_{ref} - (1 - \alpha) \times PAR_{tr}}{PAR_{in}}$$

*Where $PAR_{in}$ is incoming photosynthetic active radiation (PAR), $PAR_{ref}$ is reflected PAR, $\alpha$ is PAR albedo of the soil, and $PAR_{tr}$ is PAR transmitted through the vegetation.*

The reviewer is correct here and given reviewer 2 and 3 also highlighted this point, it prompted us to re-check our analysis of fPAR and APAR.

Initially the reason for omitting reflected PAR was due to fPAR values often being negative in the understory in the late dry season. This was most likely due to the lack of vegetation in the understory in the late dry season around some of the towers, which caused incoming PAR below the understory to be almost equal to that of PAR above the understory. For one tower, PAR below the understory was higher than PAR above the understory, which is a result of

the heterogeneous nature of the savanna ecosystem at these point scales. By omitting this tower from the analysis during the late dry season, negative fPAR values no longer occurred in the understory. This data was then used to calculate APAR, not IPAR.

Basically, there was a bug in the code that was missed on previous checks before submission. This bug was due to an incorrect labelling of the APAR variable to an alternative version, which omitted the reflected/upwelling PAR to test the above theory about the negative fPAR values. Therefore, we incorrectly concluded that by omitting the reflected PAR, the model performed better, when in fact it was actually using the correct, reflected PAR-included APAR values.

To sum up, we are grateful for the keen eyes of all three reviewers here for picking up on this mistake before the manuscript made it further in the review process. Thankfully, the data presented are correct, they were just interpreted incorrectly on our behalf. We have made the necessary changes to equations 3-5 in the manuscript, which now include the relevant reflected PAR information, and have removed the sentence stating the reason for leaving it out in the first place.
* * *
*I do not understand how the model can overestimate the GPP? You estimate a maximum LUE based on an average LUE for Dec-Mar. Then you use scalars with a value of between 0 and 1 to downscale the maximum LUE to a lower value. But since maximum LUE is based on the same time series of GPP as you use for the evaluation, it should not be possible for modelled GPP to be overestimated. Or did I misunderstand something? Please clarify.*

We are a little unsure as to what the reviewer is referring to with this statement, if it is one aspect of the text/figure or if it is our general approach to our research question. However, to answer this query at a general level, the model can most definitely overestimate GPP (or underestimate it) as LUE is not the only input to the GPP model. APAR is also an input, which in the case of the savannas is often overestimated during the transition periods between wet and dry seasons (i.e., Kanniah et al., 2009, Whitley et al., 2011). Meteorology also drives the down-regulation of maximum LUE to daily LUE variability, so although we obtained maximum (peak) LUE from our GPP estimates, the application of this down-regulation process means the two parameters are no longer directly related. Therefore, by using APAR and LUE in the model, GPP can be over- or underestimated. This is why we chose to test whether including phenology information would improve the model's ability to capture flux tower GPP, given this savanna ecosystem displays such a distinct boom-bust seasonal phenology.
* * *
**Specific comments:**
*L11, it sounds like all grass in savannas is C4 species, which is absolutely not the case. Please just rephrase a bit.*

The sentence has been changed to read "…, whereas the grasses more commonly use the C4 pathway, …"
* * *
*P6 L1 Please describe very shortly the partitioning method used. Was it based on a light response curve or night time NEE-temperature curves?*

We used a u* filter and artificial neural network approach, with soil water, soil temperature, air temperature and EVI as the main model drivers, to determine respiration (R), assuming all night time NEE was R. This was extrapolated to the daytime and GPP was calculated as the difference between R and NEE. Further information about this process can be found in Beringer et al. (2016), also an article in the special issue our manuscript is a part of. Hence, we have added this short description to P6, L1 and direct the reader to Beringer et al. (2016) for further information.
* * *
*Generally in the method section there are very many technical details. These are nice to have, but I think they could be moved to supplementary material to ease the reading of the manuscript. But, it is ok the way it is now as well, it is just a suggestion.*

We have revised the manuscript and reduced the wordiness of the methods section from 3300 words to 2885 words. In particular, we divided section 2.4 into three new sections, to more explicitly identify the phenocam image processing, the radiation data processing and the LAI and biomass measurements. While the use of supplementary material was suggested for this section to reduce its wordiness, we felt the processes discussed in each section were important enough to remain in the main body of the paper, so we chose to reduce wordiness where possible instead. For some of the additional analyses suggested by the reviewers, we have included tem as supplementary material to ensure we did not add to our shortened methods section.
* * *
*P9L18 APAR is in MJ **d-1**.*

We have fixed this in the text.
* * *
*P5 please indicate the study period of the EC measurements, and other measurements by the way.*

The study period was from 12$^{th}$ December 2012 to 14$^{th}$ October 2014, for all measurements. We will add this to the text.
* * *
*P9 L24 Why is n=8? In the figures it looks like the measurements started in January 2013, which would men n=7?*

N=8 because it includes the months of Dec through to Mar (inclusive), which each occur twice during the study period. We stated this in the text on P9, L22, but have made it clearer by also stating the timeframe of the study (n=8, across two years)
* * *
*P9 L22 Why did you bin the LUE to months, this does not necessarily give the best indicator of maximum LUE. I would say that better would be to use a running mean for the estimates of*

*seasonal dynamics in LUE, and then use the maximum value. Why should the average of 3 months give the best estimate for a maximum?*

We binned LUE by month and termed it peak LUE, rather than maximum LUE, because true maximum LUE is not easy to obtain from EC measurements. What we wanted to get at was a representative maximum LUE that was obtained during conditions that were not limiting to growth. A similar approach was used by Kanniah et al. (2009), so we intended to mirror their approach in terms of calculating a maximal LUE estimate from EC measurements. To reduce this confusion cause by our use of the term 'peak', we have changed back to 'maximum', but have clarified that for our study, $LUE_{max}$ is the general maximum light use efficiency during the wet season.

To address the reviewer's suggestion about calculation of $LUE_{max}$ using a running mean approach, we revisited our calculation process of LUE and found that FIPAR was used instead of FAPAR in the calculation of LUE. As such, Figure 3 should look like Figure 3.2 instead (below), which changes the $LUE_{max}$ values slightly, based on the Dec-Mar averaging approach. This is likely the result of the confusion identified by the reviewers about our use of FIPAR instead of FAPAR throughout the manuscript. Using the now correct LUE timeseries, we applied a 30-day running mean approach, as suggested by reviewer 1, to calculate an alternative $LUE_{max}$. This approach produced Figure 3.3. While it was a good idea to calculate $LUE_{max}$ using a running mean approach, we feel it produced unrealistically high values, particularly for the understory (4.59 g C $MJ^{-1}$ $APAR^{-1}$), when compared against other $LUE_{max}$ values reported for savanna ecosystems (i.e. 0.33 to 3.5 g C $MJ^{-1}$ $APAR^{-1}$). In addition, the running mean approach also gave a higher $LUE_{max}$ value for the overstory than that for the savanna ecosystem. The ecosystem $LUE_{max}$ should reflect a combination of overstory and understory LUE, and as such, should be at least slightly higher than the overstory. Given this, we feel our Dec-Mar averaging approach gave the most realistic $LUE_{max}$ values.

[Figure]

Figure 3: Original LUE figure displayed in manuscript, where a) Ecosystem, b) Overstory, and c) Understory.

[Figure]

Figure 3.2: Updated figure to be included in resubmission. LUE is calculated using APAR, not FIPAR.

[Figure]

Figure 3.3: LUEmax calculated from a 30-day running mean approach. Shading indicates daily LUE values.
* * *
*P16 L 17 Why did you use GCC as a proxy for FAPAR, and not as a scalar for LUE? There is strong seasonal variability in LUE depending on phenology of the vegetation, so I would think that it is more realistic to use the phenology as a direct scalar on LUE.*

We used GCC as a proxy for fPAR because the high values of fPAR in the transition periods were what we believed to be the source of the error in the model. LUE reduces rapidly from Feb to May, which is more characteristic of the phenology response seen in the field (i.e. Figure 3). Given this, the LUE was more indicative of phenology driven GPP than APAR, so was less likely to be the source of the error in the model than APAR. In addition, we calculated maximum LUE from the GPP data, which was then downregulated with VPD (or EF) and Ta to give a LUE time series. Therefore, we did not feel it appropriate to use GCC in place of LUE.
* * *
*P10 L29 I assume that the regression was not used to replace APAR, but to replace FAPAR?*

The phrasing of this sentence is misleading. It now reads "Daily EVI were regressed against site-based daily ecosystem fPAR, and the regression was used along with incoming PAR information to replace APAR in Eq. (6)."
* * *
*P12 L34 What limitations?*

The limitations refer to those mentioned in line 29 of the same paragraph. We have removed this part of the sentence, given it is largely a repeat of the information in line 29.
* * *
*P13 L4 I would not consider a R2 value of 0.09 and 0.23 a well correlated relationship. These relationshis are not well correlated just because the p-value is significant. The assumptions for testing of significance is not fulfilled; there is high auto-correlation present in eddy covariance time series, so the true N is nowhere near the observed N. For example, Desai (2014) addresses this issue using a reduced degree of freedom calculation to show that the vast majority of flux tower regression is actually over-confident.*

We agree with the reviewer here, better would be to say more broadly that the relationship was stronger for the understory than for the overstory. We will change this in the text accordingly.
* * *
*Fig 8-10. I suggest to incorporate subplots just like you did in Fig 7. Where you include a subplot with modelled GPP on the y-axis and the measured GPP on the x-axis. This really helps to see how well the models perform.*

This is a good idea. The reason we did not do it from the beginning was because we felt it made the figures too busy, so we included this information in Table 2 instead. However, we do agree that it would add to the figures, so we have included them in our resubmission.
* * *
*P15 L25-L27 Are you certain that RMSE is higher for the GCC included model (RMSE =1.43) than for the GCC and EF combined model(RMSE=1.36)? When looking at Fig 8 it does not look like RMSE can be higher. In Figure 8, it looks like the errors are much smaller; this should also be seen in the RMSE values.*

Using slightly adjusted maximum LUE values in this analysis (as per reviewer # 1's previous comment) has resulted in slight adjustments to most values in Table 2. Now, the RMSE for the LUE_GCC included model for the overstory is slightly lower (= 1.56) than for the LUE_EF_GCC model (=1.59), which is more in line with what is shown in Figure 8. In saying this, these metrics are by themselves only one indication of error in the model, which is why a combination of all three have been used as a test for model function. The RMSE approach is particularly susceptible to the cancellation effects of over- and under-estimation throughout the year, which is evident in the overstory dataset. In contrast, there is substantial variability between the RPE or the two models, with the LUE_GCC model showing far less over-prediction than the LUE_EF_GCC model (6.47 vs 16.45, respectively). This highlights the importance of considering all three metrics when assessing the effectiveness of these model runs.

Table 2 has been updated, as have the values referred to within the text from it.

*The use of GCC in the LUE model is thought to improve the GPP estimation because of the strong phenological cycle of the target. In my opinion the phenological cycle is very well represented when fAPAR is used. So the reason for using GCC must be different: replacing fAPAR measurements or testing if a "green" index (likely a proxy of a "green" fAPAR) provides a better description of photosynthesis that that of total fAPAR.*

The reviewer raises a valid point here, in that fAPAR does capture the phenological cycle reasonably well. However, it does not capture it perfectly and is particularly poor during the transition from the wet to dry season (or dry to wet). We believe this is due to the senescence of the understory grasses that changes the greenness and GPP of the savanna despite fPAR remaining high. Currently, savanna productivity models poorly capture this change (i.e., Kanniah et al., 2009, Whitley et al., 2011, Whitley et al., 2016), and we would argue it is because they do not capture the understory phenology dynamics as well as they could. Moore et al. (2016) found that the understory accounts for 1/3 of savanna GPP, which is heavily dominated by the annual grasses that show this strong phenology. When models only use the fPAR(or APAR) information, they fail to capture the transition from wet to dry (and dry to wet) and over-estimate GPP. By using the GCC information, which provides a more accurate representation of phenology when compared with APAR for this savanna, the LUE model performs better. Ma et al. (2014) also reached a similar conclusion when they used EVI to incorporate better phenology information into their GPP model.
* * *
*Cameras pointing to trees: as large part of the ROI is occupied by the background (the sky), I wonder if the observed (and reduced) variability in GCC is not related to variations in sky optical properties during the year. The relation with LAI (Fig 6b) is not helping to figure it out, as the observed relation between GCC and LAI may be spurious (i.e., LAI increase and decrease in parallel to changes in sky optical properties). To disentangle the two effects it would be useful to define some additional ROIs with sky only and analyse the difference with the tree-ROIs selected.*

This was a great idea suggested by the reviewer, so we thank them for it. We proceeded to analyse a sky-only ROI for the three overstory cameras used to generate the chromatic coordinate and excess indices. The outcome of this analysis is shown in the following figure (a), where the original Gcc timeseries for one of the towers is depicted alongside its corresponding sky-ROI timseries. At any given time, the sky GCC timseries is always less than that of the large ROI. Given this, we incorporated into our GCC image processing procedure a step where the GCC value for each pixel is excluded from the analysis if it is equal or less than the sky-ROI GCC value for that same image. This resulted in a new GCC timeseries which omitted the pixels that were sky. Thus, the new GCC timeseries (Figure a) is more representative of how the green foliage in the overstory changes over time. Given the sky threshold applied was calculated specifically for each day, the effects of changing sky conditions on each of the images has also been reduced. We applied the same technique to calculate the red (RCC) and blue (BCC) chromatic coordinates, as well as the excess coordinates. Therefore, the overstory analysis within the text has been updated, as have the relevant figures and tables. We have also included a supplementary material file outlining the

steps used in this process, in an effort to keep the length of the methods section at a minimum. We make reference to this material in section 2.4 and 3.1.
* * *
*Performances of the different GPP models (4, all LUE based) are assessed in terms of r, RMSE and RPE. However, model 1 and 2 (eq 6 and use of EF) are used in prediction while (if I got it well) model 3 (using phenocam index) is in fitting (as two parameters, m and c coefficients) are adjusted. Model 4 (using MODIS) is in between, because a relationship is tuned between EVI and fAPAR. Therefore, results are not comparable in my opinion (see the discussion at page 15).*

Each model combination is compared against flux tower-derived GPP estimates and the r, RMSE and RPE provide an indication of which model is best at capturing tower GPP. Our discussion on pg 15 discusses which of the model combinations was best at capturing tower GPP, finding that the inclusion of phenology information did the best job.
* * *
*An interesting point is that the use of the phenocam index appears to eliminate the lag between measured and modelled GPP. The reason for this could be that the total fAPAR used by the other model is the source of this lag. On the contrary GCC may represent a kind of "green" fAPAR that is more in line with photosynthesis. A dedicated section comparing phenocam indexes and fAPAR would be very useful.*

We have created plots of APAR and fPAR vs. GCC and included them in a supplementary materials file. We discuss these plots in section 3.3 and suggest that the GCC likely represents a 'green APAR' that is able to more closely track vegetation productivity over time.
* * *
**Specific comments:**

*1 L 32 r2 ranging from 0.1 to 0.2 (overstory) is much lower than that of understory but they are both indicated as "well correlated".*

Agreed, we have amended our statement in the text, as per our response to reviewer 1 above.
* * *
*3 L 23 I don't understand what is meant by "Core issues surrounding the remoteness of satellite sensors"*

Here we meant to identify that one of the limitations of satellite remote sensors is their remoteness from the ecosystems they measure. We have re-phrased the sentence to state this more clearly.
* * *
*3 L23-25 this sentence is rather obscure ("the diffuse nature of light"?). I would sug-gest to omit it and only mention that the highest temporal frequency available is one composite every 8-16 days.*

We have omitted the "diffuse nature of light" segment identified by the reviewer so the sentence reads more concisely.
* * *
*3 L 34 I don't understand "via leaf emergence and senescence". Please rephrase.*

This sentence is talking about the value of phenocams for identifying leaf-level changes, such as leaf emergence and senescence, so we have rephrased the sentence to more clearly show this. In particular, we have removed the word "via" as this seemed to be the most misleading part of the sentence.
* * *
*4 L1-3 Here you are saying that LUE models describes GPP through the relation between APAR and LUE. There is no relation, they are both used to estimate GPP.*

Here we used the word 'relation' to indicate that the two parameters were multiplied to obtain GPP. We have simplified this sentence to now reads "Phenocam data have also been used for parameterising light use efficiency (LUE) models… that describe ecosystem GPP using absorbed… APAR and plant LUE"
* * *
*Section 2.3 The final field of view of the camera could be provided.*

This has been calculated for the understory as 4 m x 2 m (horizontal x vertical) based on an object distance of 5 m, and for the overstory as 8 m x 5 m based on an object distance of 10 m. This information has been provided in section 2.3.
* * *
*Section 2.3 Can you comment on possible effects of the automatic (and variable) white balance? This can variable from measure to measure. What is the effect on calculated indexes? Few numerical simulations may help in this assessment.*

We do discuss, albeit briefly, the effects of white balance on image collection in the limitations section of our manuscript. The reviewer is correct in their assessment that white balance can vary from image to image, which is particularly more prevalent during lower sun angles i.e. dawn/dusk. By using middle of day values, the effects of white balance can be reduced. However, white balance was set to zero in our analysis, which is a limitation in that it increases the scene illumination noise in our images. However, given that we only analysed middle of day images in an environment that is highly dynamic, the phenology signal was still identifiable. This may not be the case for a less dynamic ecosystem. Migliavacca et al. (2011) discuss the uncertainty and limitations of using digital camera imagery, which we make reference to in section 3.4. We have included some additional thought on this issue in section 3.4, indicating that we believe the strong phenology of the savanna studied allowed the signals to be identified despite the potential for variable white balance.
* * *
*8 eq 16-18 Why is the reflected PAR is not used? This is fIPAR. And the resulting flux is IPAR not APAR*

Our answer to reviewer 1 about this should help to clarify this point.
* * *
*10 L 34 In which sense "predictive" is used here? Is there any validation / prediction on independent data (i.e. not used in fitting)?*

The relative 'predictive' error indicator we used in our analysis is simply a calculation of the % mean difference between two datasets. It provides an indication of the direction of change in the predicted values relative to the measured values in a relative sense. See Kanniah et al. (2009) Appendix 1 for further explanation and formulas for calculation. We have altered the sentence slightly so that it reads "…relative predictive error (RPE) to represent the percentage difference and degree of over- (+) or under- (-) estimation of the model."
* * *
*Section 3.1 It would be interesting to see the FAPAR curves along with that of the various camera-indexes*

This was a nice idea, but in the interest of balancing the additional information requests and the length of the manuscript in its current form, we think that creating and discussing an additional plot within the manuscript would make the manuscript unwieldly (there are already 10 figures and 2 tables in it). However, we have created plots of APAR and fPAR along with Gcc and included them in supplementary material, making reference to them in section 3.3.
* * *
*Figure 7. Sorry, I am not getting what the 1:1 line refers to. The two variables on the scatterplot have different units and ranges*

We can see why this would be confusing, it was meant to simply provide a guide of the deviation of the data, so we have removed the line from the resubmitted version.
* * *
**Technical corrections:**

*3 L 4 Why "cover"?*

We have removed the word 'cover' in the sentence to simply read "phenological change"
* * *
*7 L 5-7 This sentence says that it is homogeneous and it is not. It's a matter of scale. It can be rephrased.*

This is absolutely true, but the sentence in question does state this: "While the understory is largely homogenous in species distribution at the flux tower scale (i.e. >50 m), variation from one point to another does exist in the understory due to its vegetation composition." The sentence has been amended to read "…variation does exist at the smaller scale (i.e. < 5 m) in the understory due to…" to be a bit clearer on the subject.
* * *
*8 L 13 "Absorbed" instead of "used".*

We have fixed this in the revised version.
* * *
*11 L9 RCC/ExR looks like a ratio. I would suggest to use "and".*

We have fixed this in the revised version.
* * *
*13 L1 I miss the integration in this section. The title of this section could be "Relation between GPP and time series of phenocam and MODIS indexes"*

This section is about using the phenology information to improve estimates of GPP. Given this, we agree with the reviewer that the heading is misleading, therefore we have changed it to "Phenocam and MODIS phenology in relation to GPP".
* * *
*P14 L5-7 Probably not needed, already described.*

We have removed this sentence from the text.
* * *
**Reviewer #3**

*The utility of phenology information for improving GPP modeling results is an important research objective and I find the present work interesting and relevant. The paper is well written, methods are sound and results are carefully discussed. However, descriptions are generally very (too) detailed and several sections would benefit from a slightly more concise format. The structure of parts of the methods section should also be improved for improved overview, flow and clarity.*

We are pleased the reviewer enjoyed our manuscript and do agree that it is quite lengthy in parts. Given we used a rather home-made camera for our phenocams, we felt we should provide more detail about our methods. However, we have revised the manuscript and made the methods, in particular, more concise. For example, the method section was originally 3300 words in length, now it is only 2885 words.
* * *
**Some detailed and relatively minor comments:**

*1. Page 1 L32: An R2 of 0.09 – 0.23 does not constitute a well correlated relationship as I see it.*

This was also identified by reviewers 1 & 2, so we have fixed this in the text as per our response to reviewer 1.
* * *
*2. Page 2 L16: I believe fire should be capitalized as in "..2015). Fire: : :"*

Yes it should, we have fixed this in the revised manuscript.
* * *
*3. Page 3 L19: What does the A2/A3 refer to? Is this information needed here?*

The A2/A3 information refers to the sub-product of MOD17 used, as it is a combination of both GPP (A2) and NPP (A3) obtained from the Terra satellite. Given we only used the MOD17 A2 (i.e. GPP) product, we have omitted the A3 reference, but feel the A2 reference should be kept for clarity.
* * *
*4. Page 3 L20: MOD17 is mentioned to provide the most reliable means of estimating large-scale productivity. In comparison to what other products/estimates? MOD17 is known to be associated with significant uncertainty (related predominantly to the specification of the effective LUE), and I'm not convinced it will outperform other products given a full suite intercomparison.*

We agree with the reviewer here in that there are a suite of GPP model products available. However, it is out of the scope of our study to compare all products. This sentence has therefore been amended to remove the "most reliable" portion to now read "…the MODIS GPP product is widely-used means of estimating…" instead.
* * *
*5. Page 3 L23: "Core issues surrounding: : :"; Odd sentence. Suggest rewording. The full sentence structure (L23 to L28) should be rewritten for better language and clarity.*

This statement was also identified by reviewer 2, so we have fixed the sentence based on our response provided previously.
* * *
*6. Section 2 introduction (Page 4): This intro piece doesn't outline the overall methodology well and/or the sub-division of the methods sections. I would probably leave it out completely or provide a more elaborate and cohesive piece.*

The intention of this short section was to provide a brief overview/blurb of the methods before describing what was done. Given the reviewers all commented on the length of our manuscript, we have omitted it in the resubmission.
* * *
*7. Page 6 L2: I don't think that it is necessary to know the type of coding language (Python) used..*

We have removed this from the section identified.
* * *
*8. Page 6 L31: "f/stop"?*

This is a photography term that refers to the ratio of a lens' focal length to the diameter of the point where light enters the camera. It can be referred to as a focal point. We have added "(focal point)" after it in the text.

*9. Sections 2.3 and 2.4: The methods are described in great detail. I would suggest reducing the wordiness as much as possible only including the most essential elements.*

We have taken these comments on board and have reduced the length of the methods section from 3300 words to 2885 words.
* * *
*10. Section 2.4: I would include separate sub-sections for the phenocam and radiation data processing for improved flow and readability. Line 13 on page 8 could be the start of the LUE sub-section.*

This is a great suggestion and we have split the section where indicated by the reviewer.
* * *
*11. Page 7 L24-26: I feel that this information is redundant.*

We have removed this information in the re-submitted manuscript.
* * *
*12. Page 8 L22: Shouldn't leaf absorptance be considered in the APAR calculation? You are using fPAR and not fAPAR, right?*

Our response to reviewer 1 regarding this should help clarify this point.
* * *
*13. Page 8 L24-: The information on LAI collection, clumping etc is out of place. You will need a separate section on this.*

We have also separated this section into a new subsection in the methods.
* * *
*14. Page 10 L1-4: Is it valid to adopt the default MOD17 savanna values for your study site? Did you verify these against the tower observations?*

The Tmin and VPD values were previously validated for the Howard Springs site by Kanniah et al. (2009). However, we found slightly higher maximum VPD for our study period than that of Kanniah et al. (2009). Therefore, we cited the original values of Running & Zhao (2015) for our study. Given Kanniah et al. (2009) did perform a validation of earlier values of Running et al. (2006) for savannas, we have included Kanniah et al. (2009) in our citation of the section identified.
* * *
*15. Section 3.1 is very detailed and would benefit from a more concise format, if possible.*

We have shortened this section from 980 words to 772 words.
* * *
References Cited:

BERINGER, J., MCHUGH, I., HUTLEY, L. B., ISAAC, P. & KLJUN, N. 2016. Dynamic INtegrated Gap-filling and partitioning for OzFlux (DINGO). *Biogeosciences Discuss.,* 2016**,** 1-36.

KAIMAL, J. C. & FINNIGAN, J. J. 1994. *Atmospheric boundary layer flows: their structure and measurement,* New York, Oxford University Press.

KANNIAH, K. D., BERINGER, J., HUTLEY, L. B., TAPPER, N. J. & ZHU, X. 2009. Evaluation of Collections 4 and 5 of the MODIS Gross Primary Productivity product and algorithm improvement at a tropical savanna site in northern Australia. *Remote Sensing of Environment,* 113**,** 1808-1822.

MA, X., HUETE, A., YU, Q., RESTREPO-COUPE, N., BERINGER, J., HUTLEY, L. B., KANNIAH, K. D., CLEVERLY, J. & EAMUS, D. 2014. Parameterization of an ecosystem light-use-efficiency model for predicting savanna GPP using MODIS EVI. *Remote Sensing of Environment,* 154**,** 253-271.

MIGLIAVACCA, M., GALVAGNO, M., CREMONESE, E., ROSSINI, M., MERONI, M., SONNENTAG, O., COGLIATI, S., MANCA, G., DIOTRI, F., BUSETTO, L., CESCATTI, A., COLOMBO, R., FAVA, F., MORRA DI CELLA, U., PARI, E., SINISCALCO, C. & RICHARDSON, A. D. 2011. Using digital repeat photography and eddy covariance data to model grassland phenology and photosynthetic CO2 uptake. *Agricultural and Forest Meteorology,* 151**,** 1325-1337.

MOORE, C. E., BERINGER, J., EVANS, B., HUTLEY, L. B., MCHUGH, I. & TAPPER, N. J. 2016. The contribution of trees and grasses to productivity of an Australian tropical savanna. *Biogeosciences,* 13**,** 2387-2403.

WHITLEY, R., BERINGER, J., HUTLEY, L. B., ABRAMOWITZ, G., DE KAUWE, M. G., EVANS, B., HAVERD, V., LI, L., MOORE, C., RYU, Y., SCHEITER, S., SCHYMANSKI, S. J., SMITH, B., WANG, Y. P., WILLIAMS, M. & YU, Q. 2016. Challenges and opportunities in modelling savanna ecosystems. *Biogeosciences Discuss.,* 2016**,** 1-44.

[revised manuscript text omitted]

---

## Author Response (AR2)

**Author response to Editor comments for manuscript bg 2016-187**

**Editor Comments:**

**Associate Editor Decision: Publish subject to minor revisions (Editor review)** (19 Nov 2016) by Mirco Migliavacca
Comments to the Author:
Dear authors,

in the revised manuscript you addressed properly all the concerns of the reviewers. The manuscript reads quite well and the analysis robust, and it will help to support further analysis on the phenology of Savanna and in general tree/grass mixed ecosystems.

The manuscript can be published after some minor revision

>P6 L1 Please describe very shortly the partitioning method used. Was it based >on a light >response curve or night time NEE-temperature curves?
>We used a u* filter and artificial neural network approach, with soil water, soil >temperature,
>air temperature and EVI as the main model drivers, to determine respiration >(R)..

Can you please clarify which EVI was used? EVI MODIS or from field spectrometers? how was smoothed to produce daily time series (if it was)? a clarification could be useful for the readers (and at page 6 line 1 I haven't found the information).

Discussion with Reviewer #2 about automatic white balance
> However, white balance was set to zero in our analysis, which is a limitation in that >it increases the scene illumination noise in our images.

If I understand correctly the white balance was kept fixed. In this case please clarify this aspect and the comment of the reviewer is amended. Of course the use of midday data also help as stated by the authors. The use of automatic white balance can have important impact on the digital number (DN) in particular of the blue channel impacting not only the noise but the seasonality on the GCC time series.

Sincerely,
Mirco Migliavacca

**Author Response:**

We wish to thank Dr. Migliavacca for finding our responses to the reviewer comments pleasing and for recommending our manuscript for publication. To answer the first minor concern of Dr. Migliavacca, we have clarified more explicitly in the text of section 2.2 that the partitioning method used was based on night-time NEE temperature response curve fitting. In regards to the second point raised by Dr. Migliavacca, it appears there was a typo in our response to reviewer #2's comment, in that white balance was actually set to 'auto', not 'zero' (AWB stated in section 2.3). Unfortunately, the cameras were installed before our realisation of the

importance of white balance for phenocam image analysis. We discuss this limitation in section 3.4, but feel it does not substantially change the overall message of our research; that inclusion of phenology information is important to consider when modelling GPP in savanna ecosystems, which is also supported by the EVI analysis. The use of AWB is likely to have more of an impact at the hourly to daily timescales due to changing sky conditions. We smoothed our GCC data using an 8-day running mean so as to remove some of the day-to-day noise likely from the AWB setting. This also made our comparison with MODIS-EVI more appropriate. We have added some more discussion about this to section 3.4 in order to make it clear that our phenocam data has this limitation, but that we also argue our results still tell an important story about the importance of phenology for predicting savanna GPP.